# OFFLINE-TO-ONLINE REINFORCEMENT LEARNING WITH PRIORITIZED EXPERIENCE SELECTION

## ABSTRACT

Offline-to-online reinforcement learning (O2O RL) offers a promising paradigm that first pre-trains an offline policy and fine-tunes it with further online interactions. Nevertheless, the distribution shift between the offline and online phase often hinders the fine-tuning performance, sometimes even incurring performance collapse. Existing methods mitigate this by enhancing training robustness with Q-ensemble, training a density ratio estimator to balance offline and online data, etc. But they often rely on components like ensemble and have higher training costs. In this paper, we address this issue by establishing a concrete performance bound for the optimal policies between two consecutive online steps. Motivated by the theoretical insight, we propose a simple yet effective fine-tuning method, **P**rioritized **E**xperience **S**election (PES). During the online stage, PES maintains a dynamically updated priority queue containing a portion of high-return trajectories, and only selects online samples that are close to the samples in the queue for fine-tuning. In this way, the distribution shift issue can be mitigated and the fine-tuning performance can be boosted. PES is computationally efficient and compatible with numerous approaches. Experimental results on a variety of D4RL datasets show that PES can benefit different offline and O2O RL algorithms and enhance Q-value estimate. Our code is available and will be open-source.

## 1 INTRODUCTION

Online reinforcement learning (RL) (Sutton & Barto, 1999; François-Lavet et al., 2018) presents a paradigm that the agent learns an optimal policy by interacting with the environment. However, this *trial-and-error* manner also imposes inherent risks of high costs or even danger. Offline RL (Levine et al., 2020; Prudencio et al., 2023), instead, learns the optimal policy from a previously collected dataset, which could be sourced from historical data, expert knowledge, or behavior policies. Such learning paradigm is promising since it eliminates the need for interacting with the environment. Nevertheless, the performance of the offline RL algorithm often suffers from the size and quality of the underlying static dataset, e.g., learning with a small dataset with poor quality makes it challenging to learn superior policies. To leverage the advantages of both online RL and offline RL, the Offline-to-Online (O2O) RL paradigm (Xie et al., 2021; Ball et al., 2023; Wagenmaker & Pacchiano, 2023) has been explored, where the agent is first pre-trained on the offline dataset, and then further fine-tuned through online interactions with the environment. While this pre-training + fine-tuning paradigm is widely used and proved effective in computer vision (Dosovitskiy et al., 2020; Radford et al., 2021) and natural language processing (Devlin et al., 2018; Liu et al., 2019b; Brown et al., 2020; Hu et al., 2021), its effectiveness in RL is generally not as promising. Especially, during the fine-tuning phase, the "unlearning" phenomenon (Nakamoto et al., 2024; Nair et al., 2020; Uchendu et al., 2023) may occur, which means that the policy improvement is slow, or there might be a performance drop at the beginning of the fine-tuning phase. One reason for this phenomenon is the *distribution shift* between offline and online stages (Lee et al., 2022; Uchendu et al., 2023; Nair et al., 2020; Wen et al., 2023), i.e., the agent encounters unseen state-action pairs during online interaction. Due to extrapolation error (Fujimoto et al., 2019; Kumar et al., 2019) and conservatism in value function (Kumar et al., 2020; Lyu et al., 2022b), the agent cannot provide a good Q-value estimate for online samples. There are many attempts to address the distribution shift issue. For example, Wen et al. (Wen et al., 2023) leverage Q-ensemble and robustness regularization to smooth the Q-function for policy fine-tuning. However, ensemble method introduces extra computational

burden. Lee et al. (Lee et al., 2022) use balanced replay to select near-on-policy samples for fine-tuning. However, one needs to train a density ratio estimator, which increases the complexity of training. It necessities to develop a general and effective method for better policy fine-tuning.

In this paper, we propose a simple yet effective approach for online fine-tuning, **P**rioritized **E**xperience **S**election, namely PES. We begin with the theoretical insight that, at the beginning of the online phase, one should only use online transitions that do not deviate far from the visited transitions to ensure smooth policy transfer. To further guarantee fast policy adaptation, we only select good online transitions to fine-tune the policy. To that end, we maintain *a priority queue* containing a portion of high-return trajectories encountered before during the online phase, and only select online samples that are close to the samples in the queue for further fine-tuning. We determine whether the online transition is close to the queue by searching its $k$-nearest neighbors in the queue and measure their average deviations. We admit the transition if the calculated deviation is small and vice versa. We leverage the KD Tree (Bentley, 1975) for efficient implementation. Meanwhile, we dynamically update the priority queue to ensure that the samples are of high quality in the queue. In this way, we make sure that the samples used for fine-tuning stay close to the previously encountered samples, thus mitigating the distribution shift. Moreover, since the queue only contains high-return trajectories, we also ensure that good online samples are used for fine-tuning, thereby improving the sample efficiency. To further tackle the underlying over-conservative issue due to partial sample selection, we adapt the selection threshold throughout the online phase to ensure data diversity.

PES is general and can be seamlessly integrated into different offline and O2O RL algorithms for efficient online fine-tuning. Experimental results on various D4RL (Fu et al., 2020) datasets demonstrate that PES can significantly benefit offline and O2O RL algorithms and mitigate distribution shift. To ensure reproducibility, we provide the code in the supplementary materials.

## 2 BACKGROUND

We consider a Markov Decision Process (MDP) (Puterman, 1990) that can be specified by a tuple $\langle \mathcal{S}, \mathcal{A}, p, r, \rho, \gamma \rangle$, where $\mathcal{S}$ and $\mathcal{A}$ are the state space and action space, respectively, $p : \mathcal{S} \times \mathcal{A} \to \mathcal{S}$ is the transition dynamics, $r : \mathcal{S} \times \mathcal{A} \to \mathbb{R}$ is the reward function, $\rho$ is the initial state distribution, and $\gamma \in [0, 1)$ is the discount factor. The goal of reinforcement learning (RL) is to obtain a policy $\pi_\theta$ which maximizes the following object function: $\eta(\theta) = \mathbb{E}_{\pi_\theta} \left[ \sum_{t=0}^{\infty} \gamma^t r(s_t, a_t) | s_0 \sim \rho \right]$. In the context of offline RL, the agent is only accessible to a static dataset: $\mathcal{D} = \{(s_i, a_i, r_i, s_{i+1})\}_{i=1}^{N}$. Since the dataset cannot cover the entire state-action space, training solely on it will constrain the agent's performance. To further improve the performance of offline RL agents without incurring excessive costs and risks, offline-to-online RL aims to fine-tune offline-trained agents with minimal online interactions. Samples collected online are stored in $\mathcal{D}_{\text{online}}$ and training samples are drawn from $\mathcal{D} \cup \mathcal{D}_{\text{online}}$ for fine-tuning.

## 3 METHODOLOGY

### 3.1 A MOTIVATING EXAMPLE

Offline-to-online RL suffers from distribution shift during the online phase, which hinders the pretrained policy from achieving higher returns. Nevertheless, we argue that the distribution shift issue can be effectively alleviated when leveraging the sample selection approach for filtering fine-tuning data, even if the distribution shift is severe. We provide a motivating example to illustrate this point.

We choose IQL (Kostrikov et al., 2022), a popular offline RL algorithm for experiments. We first pre-train IQL on `halfcheetah-medium-v2` dataset for 1M gradient steps. To simulate a severe distribution shift, we use `halfcheetah-expert-v2` dataset for fine-tuning. As shown in Figure 1 (left), these two datasets exhibit distinct state-action distributions. We consider two approaches for fine-tuning, (a) directly fine-tuning using samples from the `halfcheetah-expert-v2` dataset, tagged as *IQL-Expert*; (b) we construct a priority queue and initialize it with top-10 trajectories in the `halfcheetah-medium-v2` dataset, and then use `halfcheetah-expert-v2` dataset to fine-tune IQL with the sample selection mechanism, i.e., ignoring samples that deviate far from the queue. We denote this variant as *IQL-PES*.

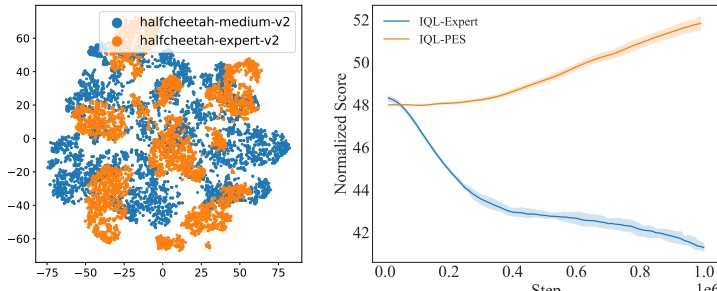

Figure 1: **Left:** Visualization results of state-action distributions of *halfcheetah-medium-v2* and *halfcheetah-expert-v2* datasets using t-SNE (Van der Maaten & Hinton, 2008). **Right:** The fine-tuning performance of IQL-Expert and IQL-PES. The shaded region denotes the standard deviation.

We present the performance comparison of IQL-Expert and IQL-PES in Figure 1 (right), where we observe a significant performance decrease for IQL-Expert during the fine-tuning stage, indicating that even when expert samples are used for fine-tuning, the performance can decline due to significant distribution shift. However, IQL-PES demonstrates robustness, and incurs stable performance improvement despite the distribution shift. The experience selection mechanism allows IQL-PES to choose expert samples with minimal distribution shift for fine-tuning. This toy example sheds light on the necessity of selecting proper and useful data during the fine-tuning phase.

### 3.2 OFFLINE-TO-ONLINE RL WITH PRIORITIZED EXPERIENCE SELECTION

For a typical online fine-tuning process, the agent collects transitions in the environment, and adds them to the online buffer $\mathcal{D}_{\text{online}}$ to fine-tune the pre-trained policy $\pi$. However, a direct fine-tuning with these samples may incur slow performance improvement or even performance drop due to distribution shift (as illustrated in the motivating example). Specifically, we argue that the data distribution or the empirical MDP (as defined in Definition 3.1) in the replay buffer between two consecutive online steps should be similar to ensure smooth policy transfer, guaranteed by theorem 3.1.

**Definition 3.1** (empirical MDP). *The empirical MDP of the replay buffer is defined by the tuple $(S, A, r, \hat{p}, \hat{\rho}, \gamma)$, where $S$, $A$, $r$ and $\gamma$ are the same as the original MDP. $\hat{\rho}$ is the state distribution of the buffer. The buffer transition dynamics $\hat{p}$ is defined as:*

$$\hat{p}(s'|s,a) = \begin{cases} \frac{\sum_{\mathcal{D} \cup \mathcal{D}_{\text{online}}} \mathbb{1}(s,a,s')}{\sum_{\mathcal{D} \cup \mathcal{D}_{\text{online}}} \mathbb{1}(s,a)}, & \text{if } (s,a,s') \in \mathcal{D} \cup \mathcal{D}_{\text{online}}, \\ 0, & \text{otherwise}, \end{cases} \quad (1)$$

*where $\mathbb{1}(\cdot)$ is the indicator function.*

**Remark:** Intuitively, $\hat{p}$ only accounts for transitions in the replay buffer; we set the probability for transitions not in the buffer to be zero.

**Theorem 3.1.** *Let $M$ be the true MDP, $\widehat{M}_t$ and $\widehat{M}_{t+1}$ be the two empirical finite-horizon MDPs between two consecutive steps $t$ and $t+1$, then the performance discrepancy between their optimal policy in the true MDP $\eta_M(\pi^\star_{\widehat{M}_t})$ and $\eta_M(\pi^\star_{\widehat{M}_{t+1}})$ can be bounded by:*

$$\left| \eta_M(\pi^\star_{\widehat{M}_t}) - \eta_M(\pi^\star_{\widehat{M}_{t+1}}) \right| \leq \frac{2(r_{\max} + \gamma V_{\max})(1 - \gamma^H)}{1 - \gamma} D_{TV}\left(p_{\widehat{M}_t}, p_{\widehat{M}_{t+1}}\right)$$
$$+ \frac{r_{\max}}{1 - \gamma}\left(D_{TV}(p_M, p_{\widehat{M}_t}) + D_{TV}(p_M, p_{\widehat{M}_{t+1}})\right).$$

*where $D_{TV}$ is the total variance distance, $r_{\max}$ and $V_{\max}$ represent the maximum value of the reward function and value function, and $H$ is the maximum MDP horizon.*

The proof is deferred to Appendix A. Theorem 3.1 suggests that if the data distribution in the replay buffer between two consecutive steps can evolve smoothly, such that the difference between the two

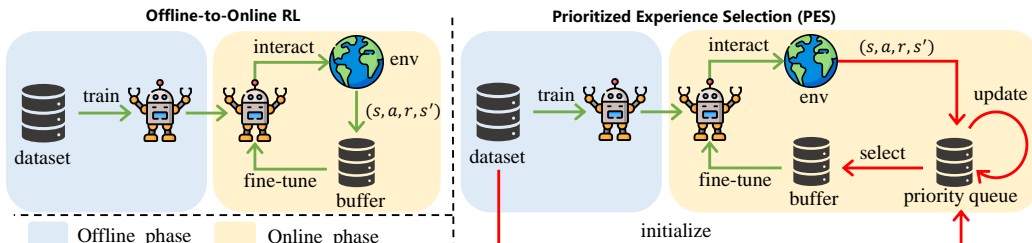

Figure 2: **Left:** Framework of offline-to-online RL. **Right:** Learning process of PES. The key difference is that PES uses the priority queue for online experience selection (highlighted in red).

estimated MDPs is small, then the learned policy can avoid the abrupt performance drop, facilitating a smooth policy transfer. Motivated by this theoretical insight, one can simply add online samples that are similar to $\mathcal{D} \cup \mathcal{D}_{\text{online}}$ to $\mathcal{D}_{\text{online}}$. However, we argue that such an approach is not ideal. On one hand, the replay buffer may contain sub-optimal trajectories with low returns. It is possible that online samples with low-quality would be admitted and used for fine-tuning, resulting in slow policy evolution. On the other hand, since the size of the replay buffer can often be large (containing more than 1M transitions), assessing the similarity between the online samples and the buffer transitions can be a heavy computational overhead. To avoid these drawbacks, we propose *prioritized experience selection* (PES), which maintains a priority queue that contains visited high-return trajectories and only favours online samples that are similar to samples in the queue. In this way, the distribution shift is alleviated and the sample efficiency can be boosted (since only high-quality samples are selected), the computational load is also significantly reduced as the priority queue holds much fewer samples compared to the entire replay buffer. Meanwhile, the queue is dynamically updated such that we can consistently include high-quality samples for higher sample efficiency. Note that the agent's exploration ability may be reduced since PES filters out proportion of online samples. To tackle this issue, we gradually loose the selection threshold throughout the fine-tuning process to ensure data diversity. This is rational since the "unlearning" phenomenon primarily occurs in the beginning of the fine-tuning phase, and we should admit more diverse online samples in the later fine-tuning stage to encourage exploration.

The core idea of PES is demonstrated in Figure 2. Compared with the previous offline-to-online RL pipeline, the main difference is that PES maintains an additional priority queue for online transition selection. The overall procedure of PES can be divided into the following steps:

**Step 1: Offline Pre-Training.** Given an offline dataset $\mathcal{D}$, we first learn a policy $\pi$ via offline RL algorithms. Since PES is orthogonal to algorithmic designs, we can adopt a variety of offline RL algorithms, such as CQL (Kumar et al., 2020), IQL (Kostrikov et al., 2022), etc.

**Step2: Constructing the Priority Queue.** After offline pre-training, we first construct a void priority queue $\mathcal{Q}$ with capacity $N$ (i.e., how many trajectories can the queue $\mathcal{Q}$ hold) to store high-quality trajectories. We sort the trajectories in $\mathcal{D}$ by their returns and push the top $N$ trajectories with the highest returns into $\mathcal{Q}$. Note that this step only aims to initialize the priority queue.

**Step 3: Prioritized Experience Selection.** This step is the core contribution of PES. For an online sample, PES evaluates its similarity to the experiences in the priority queue $\mathcal{Q}$ and adds those with high similarities to $\mathcal{D}_{\text{online}}$. By doing so, PES ensures that the samples in $\mathcal{D}_{\text{online}}$ are of high-quality, which in principle should benefit policy improvement. Moreover, to prevent the policy being overly conservative, we also *continuously update $\mathcal{Q}$ during the fine-tuning process*, as demonstrated in **Step 5**. The remaining issue is how to measure the similarity between online samples and those in $\mathcal{Q}$. One can train neural networks to fulfill that (e.g., train a classifier to determine whether the online sample belongs to the queue). But it brings heavy training costs and may suffer from training instability. We resort to the *k-nearest neighbor distance in the state-action space* as the similarity metric. Given an online sample $(s, a)$, we measure the distance between $(s, a)$ and its $k$-nearest neighbors in $\mathcal{Q}$:

$$d(s,a) = \frac{1}{k} \sum_{i=1}^{k} \left\| (s \oplus a) - (s \oplus a)^{i,\mathcal{Q}} \right\|_2 \qquad (2)$$

where $\oplus$ is the vector concatenation operator, and $(s \oplus a)^{i,\mathcal{Q}}$ is the $i$-th nearest neighbor of $(s, a)$ in the priority queue $\mathcal{Q}$, $i \in \{1, \ldots, k\}$. We then specify a selection threshold $\epsilon$. If $d(s,a)$ is

smaller than $\epsilon$, then we admit the sample $(s, a)$ and add it to $\mathcal{D}_{\text{online}}$. It is vital to decide the threshold $\epsilon$. Simply setting $\epsilon$ as a constant is not preferred since the scale of states and actions can significantly differ among different datasets. For flexibility and scalability, **we set the threshold $\epsilon$ as the maximum $k$-nearest neighbor distance of any $(s, a) \in \mathcal{D}$ against other samples in the offline dataset $\mathcal{D}$ (i.e., $\mathcal{D} \backslash \{(s,a)\}$) and gradually loose $\epsilon$ to encourage exploration:**

$$\epsilon = (1 + \alpha \cdot \frac{t}{T}) \cdot \max_{(s,a) \in \mathcal{D}} \left( \frac{1}{k} \sum_{i=1}^{k} \left\| (s \oplus a) - (s \oplus a)^{i, \mathcal{D} \backslash \{(s,a)\}} \right\|_2 \right) \tag{3}$$

where $t$ is the current fine-tuning steps, $T$ is the total steps, and $\alpha$ is a tunable hyperparameter. A larger $\alpha$ means more online samples will be added to $\mathcal{D}_{\text{online}}$ in the later online stage. We employ KD Tree (Bentley, 1975) for efficiently calculating the $k$-nearest neighbor distance. Consequently, PES only consumes a minor extra computation burden over the base algorithm.

**Step 4: Online Fine-Tuning.** During online fine-tuning, we set a sampling coefficient $\eta \in [0, 1]$, and draw a proportion of $\eta \mathcal{B}$ samples from offline dataset $\mathcal{D}$, and $(1 - \eta)\mathcal{B}$ samples from the online buffer $\mathcal{D}_{\text{online}}$, given a batch size $\mathcal{B}$. We then use these samples to fine-tune the algorithm.

**Step 5: Updating the Priority Queue.** The priority queue $\mathcal{Q}$ always stores the top $N$ trajectories with the highest returns. Since it is possible to gather high-return trajectories during online interactions, we need to maintain $\mathcal{Q}$ to reflect any new, higher-return trajectory. If the return of an online trajectory is higher than that of the trajectory with the lowest return in $\mathcal{Q}$, we pop the lowest-return trajectory and add the new trajectory to $\mathcal{Q}$. Then, the queue is sorted based on the return.

The full pseudo-code of PES is deferred to Appendix C. Furthermore, we also present some theoretical backups for PES's ability to select high-quality online samples in Appendix B.

We note that PES enjoys the following advantages: **(a)** *Compatibility with existing algorithms*: Since PES only involves online sample selection and is independent of the specific algorithmic design, PES can be seamlessly integrated into a variety of offline RL and offline-to-online RL algorithms. This flexibility allows for the enhancement of existing methods without the need to alter their core designs; **(b)** *Slight additional training costs*: PES leverages an unsupervised learning method, KD tree, to measure the distance between the online sample and the transitions in the priority queue, which is quite efficient and it does not introduce much additional training costs, as shown in Appendix G.

## 4 EXPERIMENT

In this section, we evaluate the effectiveness of PES by conducting experiments on various D4RL datasets. We first integrate PES into IQL (Kostrikov et al., 2022) in Section 4.1 and compare it with some recent baselines. In Section 4.2, we combine PES with more offline and O2O RL algorithms to examine its versatility. We further show that PES can mitigate distribution shift to benefit Q-value estimate in Section 4.3, and conduct ablation studies in Section 4.4. Lastly, we test the hyperparameter sensitivity of PES in Section 4.5.

### 4.1 MAIN RESULTS

In this part, we compare PES with other online fine-tuning methods. We adopt IQL (Kostrikov et al., 2022), a widely used offline RL algorithm as our base algorithm for PES, giving rise to IQL-PES. We choose the popular D4RL (Fu et al., 2020) benchmark for experimental evaluations. We consider 3 tasks (`halfcheetah`, `hopper`, `walker2d`), with 3 types of datasets (`random`, `medium`, `medium-replay`) for each of the task, from the MuJoCo "-v2" datasets in D4RL benchmark. We additionally choose 6 "-v0" datasets from Antmaze domain with different map sizes (`umaze`, `medium`, `large`), resulting in a total of 15 datasets for experiments. In the offline pre-training stage, we run IQL for 1M gradient steps on each dataset[1]. In the online fine-tuning stage, we transfer parameters trained offline to online stage and apply PES to IQL. All experiments run for 1M environmental steps.

---

[1] Except AWAC (Nair et al., 2020) where we follow its original training process.

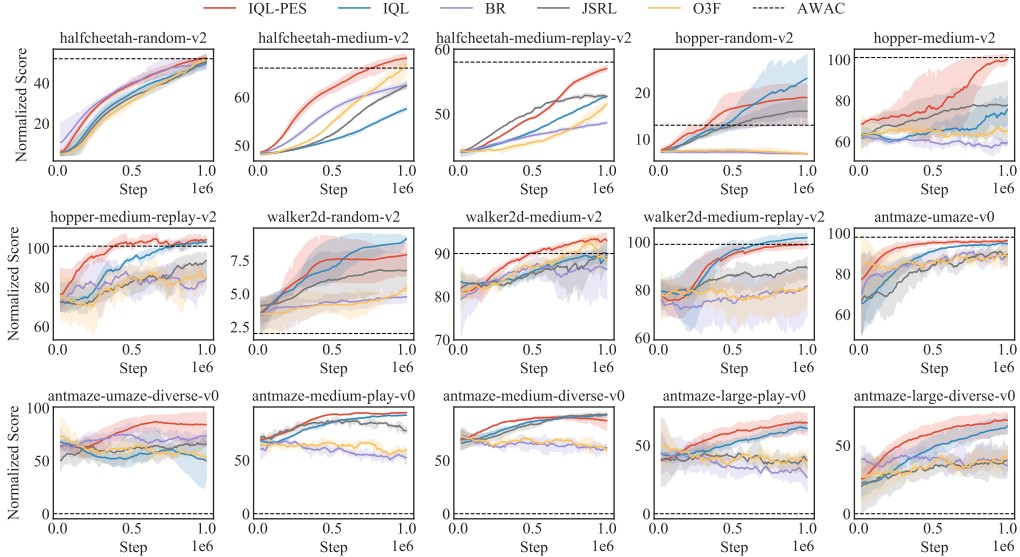

Figure 3: Online stage learning curves of IQL-PES and other baselines on D4RL datasets. The solid line is the average normalized score, and the shaded area represents 95% confidence interval. The dashed lines show the final performance of AWAC after 1M steps of fine-tuning.

**Baselines.** We consider the following baselines: *(i)* **AWAC** (Nair et al., 2020): an approach combining dynamic programming with maximum likelihood policy updates via advantage-weighted actor-critic for offline-to-online learning. *(ii)* **IQL** (Kostrikov et al., 2022), an offline RL algorithm that also trains the policy via advantage weighting. *(iii)* **BR** (Balanced Replay (Lee et al., 2022)), which selects relevant, near-on-policy offline samples for fine-tuning. The difference between PES and BR lies in that PES selects online samples. *(iv)* **JSRL** (Jump Start RL (Uchendu et al., 2023)), which utilizes a guide-policy for the rollout of the first part of the trajectory, and an exploration-policy for the rest part of the trajectory. *(v)* **O3F** (Mark et al., 2022): an optimistic action selection mechanism which encourages exploration by taking actions with higher expected Q-value. For a fair comparison, we employ IQL as the base algorithm for **BR** and **O3F** (i.e., no Q-ensemble is adopted as in their original papers). All algorithms are run across 5 varied random seeds.

**Experimental Results.** We summarize in Figure 3 the learning curves of IQL-PES and the above baselines in the online stage, where we report the normalized score for both MuJoCo tasks and Antmaze tasks. It can be found that PES significantly boosts the fine-tuning performance of IQL, and IQL-PES achieves the highest final performance in **10** out of the 15 datasets among all methods. Moreover, thanks to the online sample selection mechanism, the "unlearning" phenomenon at the beginning of online stage is mitigated (i.e., no abrupt performance degradation occurs in PES). These clearly show the effectiveness of our method. We observe that on random datasets like `hopper-random-v2`, the performance of IQL-PES is slightly inferior than the vanilla IQL. We believe this is because the data distribution of the random datasets is diverse, making the policy less affected by the distribution shift issue. Meanwhile, the offline samples in random datasets are of poor quality, and filter out online samples may potentially decrease sample efficiency. However, on datasets with a narrow data distribution, such as the medium datasets (e.g., `hopper-medium-v2`), PES can bring significant performance improvement. Notably, PES consistently beats the balanced replay approach on almost all the tasks, *emphasizing the greater significance of online sample selection over offline sample selection*. It can be observed that some fine-tuning methods, such as balanced replay and O3F, could result in a slow performance improvement, which seems to be contradictory to their original papers. We argue that the reasons are that these methods are optimistic in their action selection strategy and sampling mechanism. Such optimism could lead to more severe distribution shift, and they employ the Q-ensemble trick to enhance the robustness of Q-networks and the policy. They fail here due to the lack of ensemble Q-networks. In contrast, PES is general and does not rely on the Q-ensemble to achieve a good performance.

Table 1: Performance comparison for base algorithms w/ (denoted as "Ours") and w/o (denoted as "Base") PES on D4RL benchmark. We abbreviate "halfcheetah" as "half", "random" as "r", "medium" as "m", "medium-replay" as "m-r". We use D4RL MuJoCo "-v2" datasets and Antmaze "-v0" datasets. We report the normalized score for each dataset. All the experiments are run with 5 random seeds, and the superior normalized scores are in bold and highlighted in **green**.

| Task Name | AWAC | | PEX | | Cal-QL | | TD3-BC | | CQL | |
|---|---|---|---|---|---|---|---|---|---|---|
| | Base | Ours | Base | Ours | Base | Ours | Base | Ours | Base | Ours |
| half-r | 52.4 | **61.1** | 64.2 | **69.6** | 3.2 | **18.2** | 44.3 | **45.1** | 0.0 | **30.0** |
| half-m | 67.2 | **73.5** | **79.0** | 72.1 | 73.1 | **90.5** | 61.5 | **63.4** | 52.5 | **64.7** |
| half-m-r | 59.2 | **62.3** | 62.5 | **68.3** | **54.7** | 52.2 | 52.3 | **58.7** | **53.6** | 52.1 |
| hopper-r | 13.2 | **14.8** | 41.2 | **58.4** | 9.6 | **14.4** | 7.7 | **12.2** | **11.7** | 10.0 |
| hopper-m | 101.0 | 101.0 | 83.1 | **91.2** | 100.0 | 100.0 | 62.1 | **79.3** | 72.1 | **81.4** |
| hopper-m-r | 101.3 | **104.5** | 77.2 | **90.0** | 100.0 | 100.0 | **93.1** | 87.6 | **102.4** | 99.1 |
| walker2d-r | 2.4 | **18.6** | **24.1** | 14.7 | 6.4 | **11.3** | 5.4 | 5.4 | 6.6 | **8.4** |
| walker2d-m | **90.1** | 88.9 | **86.4** | 77.3 | 83.5 | **88.2** | 87.5 | **92.1** | 83.2 | **89.6** |
| walker2d-m-r | 98.5 | **101.3** | 94.3 | **98.1** | **95.1** | 91.9 | 88.3 | **90.2** | 97.6 | **99.8** |
| **MuJoCo total** | 585.3 | **626** | 612 | **639.7** | 527.7 | **566.7** | 502.2 | **534.0** | 479.7 | **535.1** |
| umaze | 97.3 | **99.7** | 100.0 | 100.0 | 95.9 | **99.8** | 17.4 | **33.4** | 90.8 | **99.5** |
| umaze-diverse | 0.0 | **42.6** | 79.6 | **91.7** | 64.2 | **72.3** | 0.0 | **23.7** | 77.2 | **100.0** |
| medium-diverse | 0.0 | **13.8** | **83.0** | 75.1 | 16.8 | **24.3** | 0.0 | **12.1** | 87.6 | **93.2** |
| medium-play | 0.0 | **15.6** | 88.1 | **95.3** | 17.2 | **19.0** | 0.0 | **7.4** | **93.1** | 88.1 |
| large-diverse | 0.0 | 0.0 | **63.4** | 61.0 | **1.5** | 0.0 | 0.0 | 0.0 | **76.1** | 66.3 |
| large-play | 0.0 | 0.0 | 67.2 | **80.1** | **1.1** | 0.0 | 0.0 | 0.0 | 63.3 | **69.2** |
| **Antmaze total** | 97.3 | **171.7** | 481.3 | **503.2** | 196.7 | **215.4** | 17.4 | **68.6** | 488.1 | **516.3** |
| **Total score** | 682.6 | **797.7** | 1093.3 | **1142.9** | 724.4 | **782.1** | 519.6 | **617.7** | 967.8 | **1051.4** |

## 4.2 Combining with Wider Offline and Offline-to-Online RL Algorithms

In Section 4.1, we integrate PES into IQL and demonstrate the advantages of PES. As we emphasize earlier, PES is general and can be combined with various algorithms. In this section, we aim to explore whether PES can also benefit wider off-the-shelf offline and offline-to-online RL algorithms.

**Experimental Setup.** Our goal is to show that PES is compatible to different algorithms. To that end, we integrate PES with some popular offline and offline-to-online RL algorithms, and conduct extensive experiments on D4RL benchmark. For base offline RL algorithms, we choose TD3-BC (Fujimoto & Gu, 2021) and CQL (Kumar et al., 2020) where CQL is a typical value-based offline RL algorithm that learns pessimistic value functions, and TD3-BC incorporates the behavior cloning term in the policy objective besides maximizing the Q-value. We do not make any modification to the underlying offline RL algorithms during the online fine-tuning phase, except that we adds an online sample selection process using PES. For the base offline-to-online RL algorithms, we choose AWAC (Nair et al., 2020), PEX (Zhang et al., 2023a) and Cal-QL (Nakamoto et al., 2024). For AWAC and Cal-QL, PES can be directly integrated in the online stage. As for PEX, which utilizes a fixed offline policy and a learnable online policy for policy expansion, we extend its policy set by adding another online policy. During the fine-tuning process of this new online policy, we employ PES for sample selection. We then combine these three policies to create a composite policy. We use 15 D4RL datasets for offline pre-training (1M steps) and online fine-tuning (1M steps).

**Experimental Results.** We summarize the experimental results in Table 1, which shows the final average normalized score of the base algorithms after online fine-tuning w/ and w/o PES. It can be found that for all 5 base algorithms, PES incurs significant performance boosts on both MuJoCo and Antmaze tasks, which we believe clearly verifies the effectiveness and versatility of PES. Especially, we observe that for AWAC and TD3-BC, Antmaze domain is particularly challenging, i.e., both of them can only achieve meaningful performance on the `umaze` dataset, and generally fail on other datasets. However, after applying PES, they can both learn a useful policy on challenging datasets such as `umaze-diverse` and `medium-diverse`. Notably, PES achieves a significant performance improvement for AWAC and TD3-BC in Antmaze tasks by **76.4% (97.3→171.7)** and **294%**

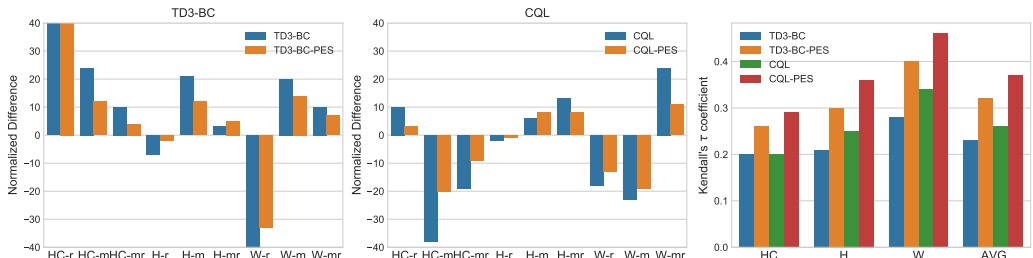

Figure 4: **Left:** Normalized difference comparison for TD3-BC and TD3-BC-PES on D4RL datasets. We abbreviate "Halfcheetah" as "HC", "Hopper" as "H", "Walker2d" as "W". **Middle:** Normalized difference comparison for CQL and CQL-PES on D4RL datasets. **Right:** Kendall's $\tau$ coefficient comparison for TD3-BC and CQL w/ and w/o PES on D4RL tasks.

**(17.4→68.6)**, respectively. Full learning curves of these algorithms are presented in Appendix H.2. We defer the full comparison results with standard deviations to Appendix H.7.

### 4.3 ENHANCEMENT OF Q-VALUE ESTIMATE

PES alleviates distribution shift by selecting online samples similar to those in the priority queue for fine-tuning, thereby yielding a more accurate Q-value estimate. In this section, we aim to empirically verify that PES can mitigate distribution shift and enhance the Q-value estimate.

**Evaluation Metrics.** Similar to (Zhang et al., 2023c), we choose the following metrics to evaluate the accuracy of the Q-value estimate: *(i)* **Normalized Difference of Q-value**: A widely used metric (Zhang et al., 2023c; Fujimoto & Gu, 2021; Chen et al., 2021; Lyu et al., 2024; Feng et al., 2024) for measuring the difference between the estimated Q-value and the true Q-value. It is computed as: $\frac{Q^{\text{estimated}} - Q^{\text{true}}}{Q^{\text{true}}}$, where $Q^{\text{estimated}}$ is the Q-value output by the Q-network and $Q^{\text{true}}$ is computed by Monte Carlo estimation (Sutton & Barto, 1999). A positive normalized difference indicates that Q-value is overestimated, and vice versa. *(ii)* **Kendall's $\tau$ coefficient** (Kendall, 1938) **over Q-value:** A metric measuring the rank correlation between two sets of variables. Given $n$ pairs of $Q^{\text{estimated}}$ and $Q^{\text{true}}$: $\{(Q_i^{\text{estimated}}, Q_i^{\text{true}})\}_{i=1}^n$, Kendall $\tau$ coefficient is computed as: $\tau = \frac{n_c - n_d}{n_0}$, where $n_c$ is the number of concordant pairs, $n_d$ is the number of discordant pairs and $n_0 = \frac{n(n-1)}{2}$. $\tau$ being closer to 1 indicates a greater positive correlation between $Q^{\text{estimated}}$ and $Q^{\text{true}}$.

**Experimental Setup.** We choose TD3-BC and CQL as the base algorithms, and evaluate the two metrics for TD3-BC, TD3-BC-PES, CQL and CQL-PES on 9 D4RL MuJoCo datasets. We calculate the normalized difference and Kendall's $\tau$ coefficient of Q-value after 1M steps of fine-tuning.

**Experimental Results.** We report the experimental results in Figure 4, where the left and middle plots show the normalized difference of Q-value for TD3-BC and CQL on 9 datasets, respectively, and the right figure displays the Kendall's $\tau$ coefficient for each task and the average Kendall's $\tau$ coefficient. It is clear that after incorporating PES, the normalized Q-value difference is reduced, and the Kendall's $\tau$ coefficient has increased by a large margin. It reveals that with the sample selection mechanism of PES, the distribution shift is alleviated and the Q-value estimate is more accurate.

### 4.4 ABLATION STUDY

In this section, we test whether varying some design choices of PES benefits or harms the performance. We mainly examine two design choices here: `Return-Prioritized Selection` and `Priority Queue Update`. For more ablation study results, we defer to Appendix H.4.

**Return-Prioritized Selection.** PES leverages a priority queue to select online samples similar to high-return trajectories. We examine the significance of this return-prioritized sample selection mechanism. In specific, we replace the priority queue with the evolving replay buffer, and select the online samples by measuring their similarity with the samples in the buffer. We conduct extensive experiments on D4RL datasets and show the results in Table 2. The results indicate that maintaining a return-prioritized queue and using it to select online samples can incur a superior performance.

Table 2: Performance comparison for IQL-PES with a priority queue or a replay buffer on various D4RL datasets.

| Task Name | Queue | Buffer |
|---|---|---|
| half-m | **68.8±3.3** | 65.2±2.9 |
| hopper-m | **100.0±1.1** | 92.6±1.7 |
| walker2d-m | 93.6±1.3 | **96.1±1.2** |
| umaze-diverse | **81.0±17.2** | 75.9±14.4 |
| medium-diverse | **88.4±5.6** | 80.6±3.2 |
| large-diverse | **66.8±6.1** | 63.9±8.1 |

Table 3: Performance comparison for IQL-PES with different updating rules (dynamically updating or fixed) on D4RL datasets.

| Task Name | Update | Fixed |
|---|---|---|
| half-m | **68.8±3.3** | 60.1±2.6 |
| hopper-m | **100.0±1.1** | 87.3±1.2 |
| walker2d-m | **93.6±1.3** | 86.1±2.6 |
| umaze-diverse | 81.0±17.2 | **84.5±11.4** |
| medium-diverse | **88.4±5.6** | 82.1±3.5 |
| large-diverse | **66.8±6.1** | 61.9±5.4 |

**Priority Queue Update.** We also examine the necessity of Step 5, i.e., always maintaining the highest-return trajectories in the priority queue. As a comparison, we fix the priority queue after initializing it as in Step 2. In this way, the priority queue only holds offline trajectories and ignores high-return online trajectories. We conduct experiments on D4RL datasets and present the results in Table 3. It is evident that keeping the priority queue fixed is an inferior choice, since it may incur conservatism and a lack of exploration.

## 4.5 Parameter Study

In this section, we examine how sensitive PES is to the introduced hyperparameters. We choose IQL as the base algorithm and conduct experiments on some Antmaze datasets. Due to space limit, we are only able to report part of our results here, and full empirical results are deferred to Appendix H.3.

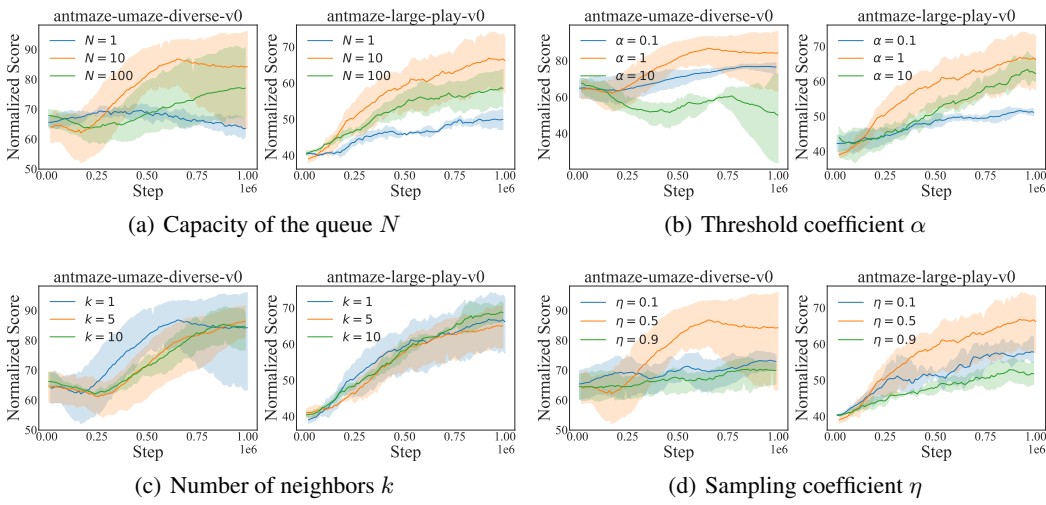

(a) Capacity of the queue $N$

(b) Threshold coefficient $\alpha$

(c) Number of neighbors $k$

(d) Sampling coefficient $\eta$

Figure 5: Parameter study of the introduced hyperparameters $N$, $\alpha$, $k$, $\eta$ in PES. The solid lines denote the average normalized scores and the shaded region captures the standard deviation.

**Capacity of the queue $N$.** $N$ represents the number of trajectories maintained in the priority queue. To check its influence, we conduct experiments by sweeping $N$ across $\{1, 10, 100\}$. The results from Figure 5(a) indicates that too small or too large $N$ is not the best choice, which is understandable since a small $N$ may reject too many online samples and a large $N$ may reject too few online samples. Fortunately, we can find a trade-off with $N = 10$. We use $N = 10$ by default in PES.

**Threshold coefficient $\alpha$.** $\alpha$ determines the threshold for sample selection and can influence the agent's exploration ability. Intuitively, PES admits more online samples in the later online stage with a large $\alpha$ and vice versa. In Figure 5(b), we vary $\alpha$ across $\{0.1, 1, 10\}$, and it turns out that $\alpha = 1$ can be a good choice.

**Number of neighbors** $k$. $k$ decides how many samples in the priority queue are used for measuring the deviation from the online sample. To see whether $k$ influences the performance of PES, we vary $k$ across $\{1, 5, 10\}$, and the results in Figure 5(c) show that PES is robust to $k$.

**Sampling coefficient** $\eta$. $\eta$ determines the proportion of samples drawn from the offline dataset $\mathcal{D}$. We vary $\eta$ across $\{0.1, 0.5, 0.9\}$, and Figure 5(d) shows that too small or too large $\eta$ will lead to performance drop, and $\eta = 0.5$ can achieve a good trade-off.

## 5 RELATED WORK

**Offline RL.** Offline RL aims to learn a constrained optimal policy with access to a static dataset. Due to the distribution shift and inability to explore (Ladosz et al., 2022; Amin et al., 2021; Liu et al., 2021; Jin et al., 2020; Lambert et al., 2022), offline RL often exhibits severe extrapolation error (Fujimoto et al., 2019). To address this issue, common strategies adopt importance sampling (Gelada & Bellemare, 2019; Liu et al., 2019a; Nachum et al., 2019; Precup et al., 2001; Sutton et al., 2016), policy constraints (Fakoor et al., 2021; Fujimoto & Gu, 2021; Ghasemipour et al., 2021; Kumar et al., 2019; Wu et al., 2019), conservative value estimation (Kumar et al., 2020; Lyu et al., 2022b; Kostrikov et al., 2021; Ma et al., 2021), uncertainty quantification (Bai et al., 2022; Wu et al., 2021; Zanette et al., 2021), and learning without querying OOD actions (Kostrikov et al., 2022; Chen et al., 2020; Wang et al., 2018; Xu et al., 2023). There are also some valuable attempts in model-based offline RL (Kidambi et al., 2020; Yu et al., 2020; Lyu et al., 2022a; Zhang et al., 2023b).

**Offline-to-Online RL.** Several studies has explored how to benefit online learning with offline data (Vecerik et al., 2017; Hester et al., 2018; Nair et al., 2018; Rajeswaran et al., 2017), which assume the datasets contain near-optimal demonstrations. However, most offline datasets are sourced from sub-optimal behavior policies and do not satisfy this assumption. A more practical manner for bridging offline and online learning phase is Offline-to-Online (O2O) RL (Lee et al., 2022; Nair et al., 2020; Wang et al., 2024; Guo et al., 2023; Lei et al., 2023), which pre-trains an offline policy and then fine-tunes it in the real environment. O2O RL also exhibits an issue of distribution shift between offline datasets and online samples. Some efforts handle this issue by selecting near-on-policy online samples (Lee et al., 2022), parameter transferring (Xie et al., 2021), policy expansion (Zhang et al., 2023a), guided exploration (Campos et al., 2021; Uchendu et al., 2023), adjusting update frequency (Zhang et al., 2023c; Feng et al., 2024), etc. There are also some researches that directly fine-tune the offline pre-trained policy without introducing additional components (Kostrikov et al., 2022; Lyu et al., 2022b; Tarasov et al., 2024a; Yang et al., 2024), but their fine-tuning performance is often limited and some of them rely on a careful hyperparameter tuning. Our work is closest to (Lee et al., 2022), but the difference lies in that PES selects *online* samples for fine-tuning by constructing a priority queue while (Lee et al., 2022) selects *offline* samples by training a density ratio estimator.

**Prioritized Experience Replay.** Our work is also related to the prioritized experience replay in RL, which prefers more essential samples in the replay buffer to benefit off-policy RL algorithms. PER (Schaul et al., 2015) prioritizes samples with larger TD-error to accelerate training, and many studies prioritize samples from different perspectives (Horgan et al., 2018; Saglam et al., 2023; Li et al., 2021; Oh et al., 2021; Pan et al., 2022). There are also studies focusing on online RL with offline demonstrations leveraging the idea of prioritized experience replay (Song et al., 2022; Vecerik et al., 2017). Our work is different from these studies in that we focus on the offline-to-online setting and we construct the priority queue for data filtering.

## 6 CONCLUSION

In this paper, we propose PES, a simple yet effective online experience selection method to handle distribution shift for offline-to-online RL. PES maintains a priority queue containing top $N$ highest-return trajectories and only selects online samples close to those in the queue for online fine-tuning. Our method is compatible with different algorithmic forms, and can incur more accurate Q-value estimate. One limitation of our work is the underlying heavy computational overhead for KNN search in high-dimensional data spaces, such as image inputs, which may harm the training efficiency. One possible solution can be using image encoders to map the original space to a hidden one, where we conduct KNN search, and we leave that for future work.

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

## A  MISSING PROOFS

In this section, we supply the missing proof for Theorem 3.1. We restate Theorem 3.1 below.

**Theorem A.1.** *Let $M$ be the true MDP, $\widehat{M}_t$ and $\widehat{M}_{t+1}$ be the two empirical finite-horizon MDPs between two consecutive steps $t$ and $t+1$, then the performance discrepancy between their optimal policy in the true MDP $\eta_M(\pi^\star_{\widehat{M}_t})$ and $\eta_M(\pi^\star_{\widehat{M}_{t+1}})$ can be bounded by:*

$$\left| \eta_M(\pi^\star_{\widehat{M}_t}) - \eta_M(\pi^\star_{\widehat{M}_{t+1}}) \right| \leq \frac{2(r_{\max} + \gamma V_{\max})(1 - \gamma^H)}{1 - \gamma} D_{TV}\left(p_{\widehat{M}_t}, p_{\widehat{M}_{t+1}}\right)$$
$$+ \frac{r_{\max}}{1 - \gamma}\left(D_{TV}(p_M, p_{\widehat{M}_t}) + D_{TV}(p_M, p_{\widehat{M}_{t+1}})\right).$$

*where $D_{TV}$ is the total variance distance, $r_{\max}$ and $V_{\max}$ represent the maximum value of the reward function and value function, and $H$ is the maximum MDP horizon.*

*Proof.*

$$\left| \eta_M(\pi^\star_{\widehat{M}_t}) - \eta_M(\pi^\star_{\widehat{M}_{t+1}}) \right|$$
$$= \left| \left(\eta_M(\pi^\star_{\widehat{M}_t}) - \eta_{\widehat{M}_t}(\pi^\star_{\widehat{M}_t})\right) + \left(\eta_{\widehat{M}_{t+1}}(\pi^\star_{\widehat{M}_{t+1}}) - \eta_M(\pi^\star_{\widehat{M}_{t+1}})\right) + \left(\eta_{\widehat{M}_t}(\pi^\star_{\widehat{M}_t}) - \eta_{\widehat{M}_{t+1}}(\pi^\star_{\widehat{M}_{t+1}})\right) \right|$$
$$\leq \underbrace{\left| \eta_M(\pi^\star_{\widehat{M}_t}) - \eta_{\widehat{M}_t}(\pi^\star_{\widehat{M}_t}) \right|}_{L_1} + \underbrace{\left| \eta_{\widehat{M}_{t+1}}(\pi^\star_{\widehat{M}_{t+1}}) - \eta_M(\pi^\star_{\widehat{M}_{t+1}}) \right|}_{L_2} + \underbrace{\left| \eta_{\widehat{M}_t}(\pi^\star_{\widehat{M}_t}) - \eta_{\widehat{M}_{t+1}}(\pi^\star_{\widehat{M}_{t+1}}) \right|}_{L_3}$$

For $L_1$, we have:

$$L_1 = \left| \eta_M(\pi^\star_{\widehat{M}_t}) - \eta_{\widehat{M}_t}(\pi^\star_{\widehat{M}_t}) \right|$$
$$= \left| \mathbb{E}_{\pi^\star}\mathbb{E}_{P_M}\left[\sum_{t=0}^{\infty}\gamma^t r(s_t, a_t)\right] - \mathbb{E}_{\pi^\star}\mathbb{E}_{P_{\widehat{M}_t}}\left[\sum_{t=0}^{\infty}\gamma^t r(s_t, a_t)\right] \right|$$
$$= \left| \sum_t \sum_{a_t} \pi(a_t|s_t)\left(p_M(\cdot|s_t, a_t) - p_{\widehat{M}_t}(\cdot|s_t, a_t)\right)\gamma^t r(s_t, a_t) \right|$$
$$\leq r_{\max} \cdot \left| \sum_t \sum_{a_t} \pi(a_t|s_t)\left|p_M(\cdot|s_t, a_t) - p_{\widehat{M}_t}(\cdot|s_t, a_t)\right|\gamma^t \right|$$
$$\leq r_{\max} \cdot \left| \sum_t \sum_{a_t} \pi(a_t|s_t) D_{TV}\left(p_M(\cdot|s_t, a_t), p_{\widehat{M}_t}(\cdot|s_t, a_t)\right)\gamma^t \right|$$
$$\leq \frac{r_{\max}}{1 - \gamma} D_{TV}(p_M, p_{\widehat{M}_t})$$

Similarly, we can get $L_2$:

$$L_2 \leq \frac{r_{\max}}{1 - \gamma} D_{TV}(p_M, p_{\widehat{M}_{t+1}})$$

For $L_3$, according to the definition of $\eta_M(\pi)$, we have $\eta_M(\pi) = V^\pi_{M,h=0}(s) := \mathbb{E}_{s\sim\rho_M}[V^\pi_M(s)]$. To get the performance bound, we can turn to calculate the value difference at horizon 0:

$$\left| V^\star_{\widehat{M}_t, h=0}(s) - V^\star_{\widehat{M}_{t+1}, h=0}(s) \right|.$$

We first consider the case at horizon $h - 1$:

$$V^{\star}_{\widehat{M}_t, h-1}(s) - V^{\star}_{\widehat{M}_{t+1}, h-1}(s)$$

$$= \max_{a \in \mathcal{A}} \left\{ \sum_{s' \in \mathcal{S}} p_{\widehat{M}_t}(s'|s,a)(r(s,a) + \gamma V^{\star}_{\widehat{M}_t, h}(s')) \right\} - \max_{a \in \mathcal{A}} \left\{ \sum_{s' \in \mathcal{S}} p_{\widehat{M}_{t+1}}(s'|s,a)(r(s,a) + \gamma V^{\star}_{\widehat{M}_{t+1}, h}(s')) \right\}$$

$$= \max_{a \in \mathcal{A}} \left\{ \sum_{s' \in \mathcal{S}} p_{\widehat{M}_{t+1}}(s'|s,a)(r(s,a) + \gamma V^{\star}_{\widehat{M}_{t+1}, h}(s')) \right\} + \max_{a \in \mathcal{A}} \left\{ \sum_{s' \in \mathcal{S}} \left( p_{\widehat{M}_t}(s'|s,a) - p_{\widehat{M}_{t+1}}(s'|s,a) \right) r(s,a) \right\}$$

$$+ \max_{a \in \mathcal{A}} \left\{ \gamma \sum_{s' \in \mathcal{S}} \left( p_{\widehat{M}_t}(s'|s,a) V^{\star}_{\widehat{M}_t, h}(s') - p_{\widehat{M}_{t+1}}(s'|s,a) V^{\star}_{\widehat{M}_{t+1}, h}(s') \right) \right\}$$

$$- \max_{a \in \mathcal{A}} \left\{ \sum_{s' \in \mathcal{S}} p_{\widehat{M}_{t+1}}(s'|s,a)(r(s,a) + \gamma V^{\star}_{\widehat{M}_{t+1}, h}(s')) \right\}$$

$$= \max_{a \in \mathcal{A}} \left\{ \sum_{s' \in \mathcal{S}} \left( p_{\widehat{M}_t}(s'|s,a) - p_{\widehat{M}_{t+1}}(s'|s,a) \right) r(s,a) \right\}$$

$$+ \max_{a \in \mathcal{A}} \left\{ \gamma \sum_{s' \in \mathcal{S}} \left( p_{\widehat{M}_t}(s'|s,a) V^{\star}_{\widehat{M}_t, h}(s') - p_{\widehat{M}_{t+1}}(s'|s,a) V^{\star}_{\widehat{M}_{t+1}, h}(s') \right) \right\}$$

$$\leq \max_{a \in \mathcal{A}} \left\{ \sum_{s' \in \mathcal{S}} \left| p_{\widehat{M}_t}(s'|s,a) - p_{\widehat{M}_{t+1}}(s'|s,a) \right| r_{\max} \right\} + \gamma \max_{a \in \mathcal{A}} \left\{ p_{\widehat{M}_{t+1}}(s'|s,a) \left( V^{\star}_{\widehat{M}_t, h} - V^{\star}_{\widehat{M}_{t+1}, h} \right) \right\}$$

$$+ \gamma \max_{a \in \mathcal{A}} \left\{ \sum_{s' \in \mathcal{S}} \left| p_{\widehat{M}_t}(s'|s,a) - p_{\widehat{M}_{t+1}}(s'|s,a) \right| V^{\star}_{\widehat{M}_t, h}(s') \right\}$$

$$\leq 2 D_{TV} \left( p_{\widehat{M}_t}, p_{\widehat{M}_{t+1}} \right) (r_{\max} + \gamma V_{\max}) + \gamma \left( V^{\star}_{\widehat{M}_t, h}(s) - V^{\star}_{\widehat{M}_{t+1}, h}(s) \right)$$

We denote $\left( V^{\star}_{\widehat{M}_t, h} - V^{\star}_{\widehat{M}_{t+1}, h} \right)$ as $a_h$, $2 D_{TV} \left( p_{\widehat{M}_t}, p_{\widehat{M}_{t+1}} \right) (r_{\max} + \gamma V_{\max})$ as $C$. Then we have:

$$a_{h-1} \leq C + \gamma a_h$$

$$\Rightarrow a_{h-1} - \frac{C}{1 - \gamma} \leq \gamma \cdot \left( a_h - \frac{C}{1 - \gamma} \right)$$

Then it is easy to have:

$$a_0 - \frac{C}{1 - \gamma} \leq \gamma^H \left( a_H - \frac{C}{1 - \gamma} \right)$$

According to the definition of value function $V_M$, $a_H = V^{\star}_{\widehat{M}_t, H} - V^{\star}_{\widehat{M}_{t+1}, H} = 0 - 0 = 0$. So we can get the upper bound:

$$\eta_{\widehat{M}_t}(\pi^{\star}_{\widehat{M}_t}) - \eta_{\widehat{M}_{t+1}}(\pi^{\star}_{\widehat{M}_{t+1}}) = V^{\star}_{\widehat{M}_t, 0} - V^{\star}_{\widehat{M}_{t+1}, 0} \leq (1 - \gamma^H) \cdot \frac{2 D_{TV}(p_{\widehat{M}_t}, p_{\widehat{M}_{t+1}})(r_{\max} + \gamma V_{\max})}{1 - \gamma}$$

Similarly, to get the lower bound, we can replace $\widehat{M}_t$ with $\widehat{M}_{t+1}$ and $\widehat{M}_{t+1}$ with $\widehat{M}_t$ in the above derivation procedure. Ultimately, we can have the performance bound:

$$\left| \eta_{\widehat{M}_t}(\pi^{\star}_{\widehat{M}_t}) - \eta_{\widehat{M}_{t+1}}(\pi^{\star}_{\widehat{M}_{t+1}}) \right| \leq \frac{2(r_{\max} + \gamma V_{\max})(1 - \gamma^H)}{1 - \gamma} D_{TV} \left( p_{\widehat{M}_t}, p_{\widehat{M}_{t+1}} \right).$$

Then we can get the objective:

$$\left| \eta_M(\pi^{\star}_{\widehat{M}_t}) - \eta_M(\pi^{\star}_{\widehat{M}_{t+1}}) \right| \leq \frac{2(r_{\max} + \gamma V_{\max})(1 - \gamma^H)}{1 - \gamma} D_{TV} \left( p_{\widehat{M}_t}, p_{\widehat{M}_{t+1}} \right)$$

$$+ \frac{r_{\max}}{1 - \gamma} \left( D_{TV}(p_M, p_{\widehat{M}_t}) + D_{TV}(p_M, p_{\widehat{M}_{t+1}}) \right).$$

That concludes the proof.

$\square$

## B THEORETICAL BACKUP FOR PES

In this part, we give some theoretical backups for PES's ability to select high-quality samples. We have the static offline dataset $\mathcal{D}$, the online buffer $\mathcal{D}_{\text{online}}$, and the priority queue $\mathcal{Q}$ in PES. We assume that the behavior policy in $\mathcal{D}$ gives $\mu$, the behavior policy in $\mathcal{D}_{\text{online}}$ is $\mu_b$, and the behavior policy in $\mathcal{Q}$ is $\mu_q$.

Firstly, we define the Lipschitz function as follows:

**Definition B.1** (Lipschitz function). *A function $f : R^m \to R^n$ is called a Lipschitz function if there exists a constant $K \geq 0$ such that:*

$$\|f(x) - f(y)\| \leq K\|x - y\| \tag{4}$$

*for any $x, y \in R^m$. $\| \cdot \|$ represents the norm, and $K$ is called Lipschitz constant.*

We assume that the reward signals, as well as the state space and action space, are bounded. To be specific, we have the following assumption:

**Assumption B.1.** *The rewards are bounded, i.e., $|r(s,a)| \leq r_{\max}, \forall s, a$. Furthermore, the state space and the action space are also bounded, i.e., $\|s\|_2 \leq C_s < \infty, \|a\|_2 \leq C_a < \infty, \forall s \in \mathcal{S}, a \in \mathcal{A}$, where $C_s, C_a$ are constants.*

The above assumption can be usually satisfied in practice, because it is less likely that we encounter boundless states or actions. The reward function is often manually written and is usually bounded. Given the above assumption, it is not difficult to derive that the $Q$ function satisfies: $|Q(s,a)| \leq \frac{r_{\max}}{1-\gamma}$, i.e., the $Q$ function is also bounded.

Denote the learned current policy as $\pi$ and the corresponding $Q$ function as $Q(s,a)$. We then further make the following assumptions about the behavior policy in the priority queue, $\mu_q$, and $Q(s,a)$.

**Assumption B.2.** *The behavior policy in the priority queue $\mathcal{Q}$, $\mu_q$, is deterministic and satisfies the Lipschitz condition with a Lipschitz constant $K_\mu$, i.e.,*

$$\|\mu_q(\cdot|s_1) - \mu_q(\cdot|s_2)\| \leq K_\mu\|s_1 - s_2\| \tag{5}$$

*for all $s_1, s_2 \in \mathcal{S}$.*

**Assumption B.3.** *The $Q$-function $Q(s,a)$ is a Lipschitz function with $K_Q$ the Lipschitz constant, i.e.,*

$$\|Q(s_1,a_1) - Q(s_2,a_2)\| \leq K_Q\|s_1 \oplus a_1 - s_2 \oplus a_2\| \tag{6}$$

*for all $(s_1,a_1), (s_2,a_2) \in \mathcal{S} \times \mathcal{A}$.*

The Lipschitz assumptions are popular and have been used in many previous RL papers (Asadi et al., 2018; Ran et al., 2023). The assumption on the $Q$ function is valid since it is bounded, and this assumption can be satisfied by properly choosing $K_Q$.

For any given online sample $(s,a)$, we follow PES and query its $k$-nearest neighbors in the priority queue $\mathcal{Q}$, and measure the distance $d$. We denote the nearest neighbors as $\{(\hat{s}_1, \hat{a}_1), \ldots, (\hat{s}_k, \hat{a}_k)\}$. If the sample resembles the samples in $\mathcal{Q}$, it is guaranteed that $d(s,a) \leq \epsilon$ in PES. We then have the following lemma.

**Lemma B.1.** *If the online sample $(s,a)$ can be admitted into the online buffer, i.e., it satisfies that its measured distance $d(s,a) \leq \epsilon$. We suppose that $(s,a)^{i,\mathcal{Q}} = (\hat{s}_i, \hat{a}_i)$ are nearest neighbors of the query sample, where $i \in \{1, \ldots, k\}$. Then, we have*

$$\hat{d}(s,a) := \|(s \oplus a) - (\hat{s}_1 \oplus \hat{a}_1)\| \leq \frac{1}{k} \sum_{i=1}^{k} \|(s \oplus a) - (s \oplus a)^{i,\mathcal{Q}}\| = d(s,a) \leq \epsilon. \tag{7}$$

*Proof.* It is easy to find that

$$d(s,a) = \frac{1}{k} \sum_{i=1}^{k} \|(s \oplus a) - (s \oplus a)^{i,\mathcal{Q}}\|$$

$$\geq \frac{1}{k} \sum_{i=1}^{k} \|(s \oplus a) - (\hat{s}_1 \oplus \hat{a}_1)\| = \|(s \oplus a) - (\hat{s}_1 \oplus \hat{a}_1)\| = \hat{d}(s,a),$$

where the inequality is due to the fact that $(\hat{s}_1, \hat{a}_1)$ are the nearest neighbor of $(s,a)$. Suppose $(\hat{s}_2, \hat{a}_2)$ are the 2-th nearest neighbor, then we have $\|(s \oplus a) - (\hat{s}_2 \oplus \hat{a}_2)\| \geq \|(s \oplus a) - (\hat{s}_1 \oplus \hat{a}_1)\|$ (otherwise, $(\hat{s}_2, \hat{a}_2)$ would become the nearest neighbor). Extending the above conclusion to other neighbors and we have the conclusion naturally. By using the fact that $d(s,a) \leq \epsilon$, we then also have $\hat{d}(s,a) \leq d(s,a) \leq \epsilon$. That completes the proof. $\square$

We now theoretically investigate whether PES is able to select high-quality samples.

**Proposition B.1.** *Suppose that Assumption B.2 and Assumption B.3 hold. For any online sample $(s,a)$, we denote its nearest neighbor in $\mathcal{Q}$ gives $(\hat{s}_1, \hat{a}_1)$, then by using PES we have*

$$\|Q(s,a) - Q(s,\mu_q)\| \leq K_Q \|(s \oplus a) - (\hat{s}_1 \oplus \hat{a}_1)\| + (1 + K_\mu)K_Q \|s - \hat{s}_1\|, \quad (8)$$

*and furthermore,*

*(a) if $(s,a)$ can be admitted, we have*

$$Q(s,a) \geq Q(s,\mu_q) - K_Q(2 + K_\mu)\epsilon. \quad (9)$$

*(b) if $(s,a)$ is rejected, then we have*

$$Q(s,a) \geq Q(s,\mu_q) - 2K_Q C_a. \quad (10)$$

*Proof.* By using Assumption B.3, we have

$$\begin{aligned}
\|Q(s,a) - Q(s,\mu_q)\| &\leq K_Q \|(s \oplus a) - (s \oplus \mu_q)\| \\
&\leq K_Q \|(s \oplus a) - (\hat{s}_1 \oplus \hat{a}_1)\| + K_Q \|(\hat{s}_1 \oplus \hat{a}_1) - (s \oplus \mu_q)\| \\
&\leq K_Q \|(s \oplus a) - (\hat{s}_1 \oplus \hat{a}_1)\| + K_Q (\|s - \hat{s}_1\| + \|\mu_q - \hat{a}_1\|) \\
&\leq K_Q \|(s \oplus a) - (\hat{s}_1 \oplus \hat{a}_1)\| + K_Q (\|s - \hat{s}_1\| + K_\mu \|s - \hat{s}_1\|) \\
&= K_Q \|(s \oplus a) - (\hat{s}_1 \oplus \hat{a}_1)\| + (1 + K_\mu)K_Q \|s - \hat{s}_1\|.
\end{aligned}$$

(a) If the sample $(s,a)$ can be admitted, by using Lemma B.1, we have

$$\hat{d}(s,a) = \|(s \oplus a) - (\hat{s}_1 \oplus \hat{a}_1)\| \leq \epsilon.$$

Meanwhile, we have

$$\|s - \hat{s}_1\| \leq \|(s \oplus a) - (\hat{s}_1 \oplus \hat{a}_1)\| \leq \epsilon.$$

By combining these results, we have

$$Q(s,a) \geq Q(s,\mu_q) - K_Q(2 + K_\mu)\epsilon. \quad (11)$$

(b) If the sample $(s,a)$ is rejected, then we have

$$\|Q(s,a) - Q(s,\mu_q)\| \leq K_Q \|(s \oplus a) - (s \oplus \mu_q)\| = K_Q \|a - \mu_q\| \leq K_Q(\|a\| + \|\mu_q\|) = 2K_Q C_a.$$

That completes the proof. $\square$

**Remark**: Proposition B.1 presents the $Q$-value deviation given the online sample $(s,a)$ and the behavior policy in the priority queue $\mu_q$. If the online sample is accepted by the PES, then we find that the expected return starting from $(s,a)$ is lower bounded by $K_Q(2 + K_\mu)\epsilon$. We can guarantee that *the selected sample can be at least as good as $(s, \mu_q(s))$*, i.e., at least as good as the behavior policy in $\mathcal{Q}$, as long as we choose a proper $\epsilon$. Moreover, if the sample is rejected, we observe that the lower bound involves $K_Q C_a$, which is a constant and $C_a$ can not be controlled. That being said, $C_a$ can be quite large. Then, it is hard to tell whether training upon $(s,a)$ can incur a good performance, and $(s,a)$ can be a quite bad sample. Therefore, we conclude that PES can theoretically guarantee that the admitted samples are of high quality.

## C  PSEUDO-CODE FOR PES

We provide the full pseudo-code for PES in Algorithm 1 to demonstrate its process. Note that one can choose the same or different RL algorithms for offline and online phases.

---

**Algorithm 1** PES: Prioritized Experience Selection for Offline-to-Online RL

---

1: **Require:** Initial Q-network $Q_\phi$, initial policy $\pi_\theta$, offline RL algorithm $\{L_{\text{offline}}^{Q_\phi}, L_{\text{offline}}^{\pi_\theta}\}$, online RL algorithm $\{L_{\text{online}}^{Q_\phi}, L_{\text{online}}^{\pi_\theta}\}$, offline dataset $\mathcal{D}$, online dataset $\mathcal{D}_{\text{online}} \leftarrow \emptyset$, priority queue $\mathcal{Q} \leftarrow \emptyset$, total offline steps $N$, total online episodes $E$, online horizon $H$
2: **for** offline step in 1 to $N$ **do**
3:     $\phi \leftarrow \phi - \lambda \nabla_\phi L_{\text{offline}}^Q(\phi), \qquad \theta \leftarrow \theta - \lambda \nabla_\theta L_{\text{offline}}^\pi(\theta)$  ◁ Step 1
4: **end for**
5: Initialize priority queue $\mathcal{Q}$ using $\mathcal{D}$  ◁ Step 2
6: Obtain the minimum return $R_{\min}$ among trajectories stored in $\mathcal{Q}$
7: Calculate the selection threshold $\epsilon$ using Equation (3).
8: **for** epoch from 1 to $E$ **do**
9:     Sample an initial state $s_0$ from state space
10:     **for** $h$ in 0 to $H - 1$ **do**
11:         Take an action $a_h \sim \pi_\theta(\cdot|S)$, observe $s_{h+1}, r_{h+1}$
12:         Calculate the $k$-nearest neighbor distance $d(S, a_h)$ using Equation (2)
13:         **if** $d(S, a_h) < \epsilon$ **then**
14:             Add $(S, a_h, r_h, s_{h+1})$ to $\mathcal{D}_{\text{online}}$  ◁ Step 3
15:         **end if**
16:         Sample a batch of transitions from $\mathcal{D} \cup \mathcal{D}_{\text{online}}$ and optimize $Q_\phi$ and $\pi_\theta$
17:         $\phi \leftarrow \phi - \lambda \nabla_\phi L_{\text{online}}^Q(\phi), \qquad \theta \leftarrow \theta - \lambda \nabla_\theta L_{\text{online}}^\pi(\theta)$  ◁ Step 4
18:     **end for**
19:     **if** $\sum_{h=0}^{H-1} r_h > R_{\min}$ **then**
20:         Update priority queue $\mathcal{Q}$ and $R_{\min}$  ◁ Step 5
21:     **end if**
22: **end for**

---

## D  DATASETS AND EVALUATION METRIC ON D4RL BENCHMARK

In this part, we provide a detailed description on the datasets we use in this paper. The offline datasets are taken directly from the D4RL (Fu et al., 2020) benchmark, which is a popular benchmark designed for evaluating offline RL algorithms.

### D.1  MUJOCO DATASETS

MuJoCo datasets are collected through interactions with continuous control tasks in Gym simulated by MuJoCo (Todorov et al., 2012). The tasks we use are `halfcheetah`, `hopper` and `walker2d`, as illustrated in Figure 6. For each task, we use the three types of datasets: *(i)* **Random**: data collected with a random policy. *(ii)* **Medium**: 1M samples collected by an early-stopped SAC policy. *(iii)* **Medium-Replay**: 1M samples from the replay buffer of the agent trained up to the performance of a medium level agent. The dataset version we use in our work is "-v2".

### D.2  ANTMAZE DATASETS

In Antmaze tasks, an 8-DOF "Ant" quadraped robot is required to reach a goal location. Antmaze tasks is more challenging than MuJoCo tasks for RL algorithms due to its sparse reward setting. There are three maze layouts contained in Antmaze tasks: `umaze`, `medium`, `large`, as shown in Figure 7. The datasets are collected in three flavors: *(i)* the robot needs to reach a specified goal from a fixed start point (`antmaze-umaze-v0`). *(ii)* the robot is required to reach a random goal from a random start point (the `diverse` datasets). *(iii)* the robot is commanded to reach specific locations from a different set of specific start locations (the `play` datasets). In our work, we use the six Antmaze datasets:

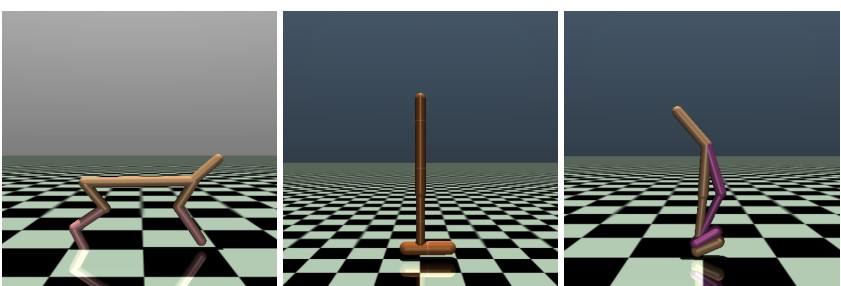

Figure 6: D4RL MuJoCo tasks. **Left:** halfcheetah, **Middle:** hopper, **Right:** walker2d.

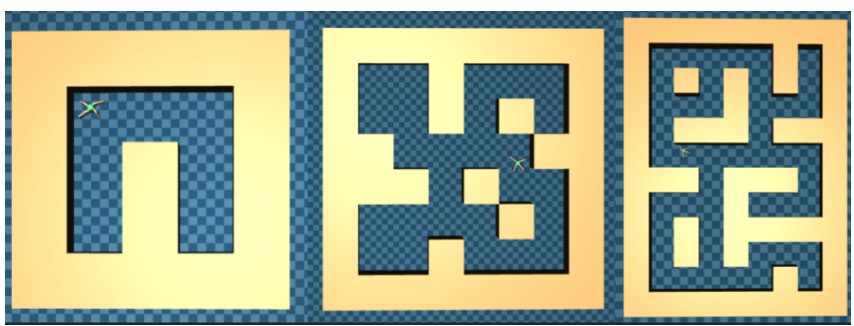

Figure 7: D4RL Antmaze tasks. **Left:** umaze, **Middle:** medium, **Right:** large.

```
antmaze-umaze, antmaze-umaze-diverse, antmaze-medium-diverse,
antmaze-medium-play, antmaze-large-diverse, antmaze-large-play.
```
The dataset version we use is "-v0".

### D.3 ADROIT DATASETS

In Adroit domain, there is a 24-DoF Shadow Hand robot required to perform several manipulation tasks. Adroit domain is quite challenging for most RL algorithms due to its sparse reward setting and insufficiency of expert demonstrations. In this work, we use the four tasks: `pen`, `hammer`, `door`, `relocate`. Each task contains three types of datasets: *(i)* **human**: several demonstrations operated by a human. *(ii)* **expert**: expert data from a fine-tuned RL policy. *(iii)* **cloned**: a 50-50 mixure of human demonstrations and rollout data from a cloned policy trained via imitation learning. The dataset version we use is "-v0".

### D.4 EVALUATION METRIC

For MuJoCo, Antmaze and Adroit tasks, we use the Normalized Score (NS) suggested by D4RL to evaluate the performance of RL algorithms. NS is computed as in Equation (12), where $J_\pi$ is the performance of the policy for evaluation, $J_{\mathrm{random}}$ is the performance of a random policy, and $J_{\mathrm{expert}}$ is the performance of an expert policy.

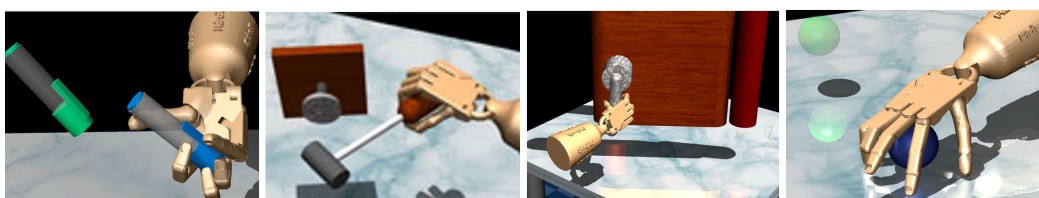

Figure 8: D4RL Adroit tasks. From left to right, pen, hammer, door, relocate.

$$NS = \frac{J_\pi - J_{\text{random}}}{J_{\text{expert}} - J_{\text{random}}} \times 100. \tag{12}$$

# E  IMPLEMENTATION DETAILS

In this part, we present the details of baseline implementations, PES implementations, and hyperparameter setup.

## E.1  BASELINE IMPLEMENTATION

Our baselines include IQL (Kostrikov et al., 2022), CQL (Kumar et al., 2020), TD3-BC (Fujimoto & Gu, 2021), AWAC (Nair et al., 2020), PEX (Zhang et al., 2023a), Cal-QL (Nakamoto et al., 2024), Balanced Replay (Lee et al., 2022), JSRL (Uchendu et al., 2023), and O3F (Mark et al., 2022). For the offline pre-training process and online fine-tuning process of IQL, CQL, AWAC and Cal-QL, we use the code from CORL[2] (Tarasov et al., 2024b), which provides reliable implementations for different offline and offline-to-online RL algorithms. For TD3-BC, since CORL only provides offline training code, we additionally implement our own online fine-tuning code. For PEX, we use the official code[3] to replicate the results in MuJoCo and Antmaze domain. For Balanced Replay, we do not follow the official code[4] that adopts the CQL-based pessimistic Q-ensemble technique. Instead, we use IQL as the base algorithm. For JSRL, the core idea is to divide the full trajectory into two parts, and utilize a guide-policy for the rollout of the first part of the trajectory, and a exploration-policy for the rest part of the trajectory. We use the offline trained policy $\pi_{\text{off}}$ as the guide-policy, and current policy $\pi_\theta$ as the exploration-policy, and use a linear scheduler that anneals from the max trajectory length to 0 for decreasing the first part of rollout by $\pi_{\text{off}}$. For the implementation of O3F, we add 10 random noise samples to each action and select the perturbed action with the highest Q-value for execution, without the use of Q-ensemble. Regarding training steps of these baselines, we set them uniformly to 1M gradient steps for offline pre-training, and 1M environmental steps for online fine-tuning.

## E.2  IMPLEMENTATION OF PES

We then provide the implementation details of PES. We implement PES upon baselines discussed above. Our major modifications are (i) we construct a priority queue where the trajectories are sorted based on their cumulative return. (ii) we select online samples based on their distances to samples in the priority queue. We do not make any other change to the base algorithms. For the implementation of (i), we utilize a three-dimensional array to store trajectory samples, with a shape of `(number of trajectories, trajectory length, sample dimension)`. To implement prioritized selection, we use the `numpy` library, i.e., `numpy.argsort()` function for sorting returns of the trajectories. For the implementation of (ii), we measure the $k$-nearest neighbor distances of online samples against samples in the queue in state-action spaces. In specific, we concatenate the state and action dimensions of samples in the queue and construct a KD Tree for efficient $k$-nearest neighbor search. We use the implementation of KD tree from `sklearn` library, i.e., `sklearn.neighbors.KDTree`. Note that we can directly get the distances when querying KD Tree.

## E.3  HYPERPARAMETER SETUP

In the main text, we conduct experiments on 9 MuJoCo datasets, 6 Antmaze datasets. We additionally include experiments on 12 Adroit datasets, yielding a total of **27** datasets. Table 4, Table 5, Table 6 present the detailed hyperparameter setup for baseline algorithms and PES on MuJoCo, Antmaze, Adroit datasets, respectively. It is worth noting that we adopt one set of hyperparameters for PES on a specific domain and keep them fixed across all runs.

---

[2]https://github.com/tinkoff-ai/CORL.git

[3]https://github.com/Haichao-Zhang/PEX.git

[4]https://github.com/shlee94/Off2OnRL.git

Table 4: Hyperparameter setup for baseline algorithms and PES on D4RL MuJoCo datasets.

|  | **Hyperparameter** | **Value** |
|---|---|---|
| Shared Configurations | Hidden layer | (256,256) |
|  | Discounted factor | 0.99 |
|  | Batch size | 256 |
|  | Critic learning rate | $3 \times 10^{-4}$ |
|  | Actor learning rate | $3 \times 10^{-4}$ |
|  | Optimizer | Adam (Kingma & Ba, 2014) |
|  | Activation function | ReLU (Agarap, 2018) |
| IQL | Value learning rate | $3 \times 10^{-4}$ |
|  | Inverse temperature $\beta$ | 3.0 |
|  | Expectile $\tau$ | 0.7 |
| CQL | Regularization coefficient $\alpha$ | 10.0 |
|  | Temperature | 1.0 |
| TD3-BC | Policy noise | 0.2 |
|  | Delay frequency | 2 |
|  | Normalization weight | 2.5 |
| Cal-QL | Regularization coefficient $\alpha$ | 10.0 |
|  | Temperature | 1.0 |
| AWAC | Lagrange coefficient $\lambda$ | 1.0 |
| PES | Search vector | $s \oplus a$ |
|  | Distance measure | Euclidean distance |
|  | Capacity of the queue $N$ | 10 |
|  | Threshold coefficient $\alpha$ | 1 |
|  | Number of neighbors $k$ | 1 |
|  | Sampling coefficient $\eta$ | 0.5 |

Table 5: Hyperparameter setup for base methods and PES on D4RL Antmaze datasets. The shared configuration is aligned with Table 4.

|  | Hyperparameter | Value |
|---|---|---|
| IQL | Value learning rate | $3 \times 10^{-4}$ |
|  | Inverse temperature $\beta$ | 10.0 |
|  | Expectile $\tau$ | 0.9 |
| CQL | Regularization coefficient $\alpha$ | 5.0 |
|  | Temperature | 1.0 |
|  | Reward scale | 10.0 |
|  | Reward bias | -5.0 |
| TD3-BC | Policy noise | 0.2 |
|  | Delay frequency | 2 |
|  | Normalization weight | 2.5 |
| Cal-QL | Regularization coefficient $\alpha$ | 10.0 |
|  | Temperature | 1.0 |
|  | Reward scale | 5.0 |
|  | Reward bias | -1.0 |
| AWAC | Lagrange coefficient $\lambda$ | 1.0 |
| PES | Search vector | $s \oplus a$ |
|  | Distance measure | Euclidean distance |
|  | Capacity of the queue $N$ | 10 |
|  | Threshold coefficient $\alpha$ | 1 |
|  | Number of neighbors $k$ | 1 |
|  | Sampling coefficient $\eta$ | 0.5 |

Table 6: Hyperparameter setup for IQL, AWAC, and PES on D4RL Adroit datasets. The shared configuration is aligned with Table 4.

|  | Hyperparameter | Value |
|---|---|---|
| IQL | Value learning rate | $3 \times 10^{-4}$ |
|  | Inverse temperature $\beta$ | 3.0 |
|  | Expectile $\tau$ | 0.8 |
| AWAC | Lagrange coefficient $\lambda$ | 1.0 |
| PES | Search vector | $s \oplus a$ |
|  | Distance measure | Euclidean distance |
|  | Capacity of the queue $N$ | 10 |
|  | Threshold coefficient $\alpha$ | 1 |
|  | Number of neighbors $k$ | 1 |
|  | Sampling coefficient $\eta$ | 0.5 |

Table 7: Average fine-tuning time cost of base algorithms w/ and w/o PES on 9 D4RL MuJoCo datasets (1M steps). "h" stands for "hour(s)" and "m" represents "minute(s)".

|       | IQL     | CQL      | TD3-BC  |
|-------|---------|----------|---------|
| Base  | 5h 51m  | 13h 21m  | 6h 06m  |
| +PES  | 6h 24m  | 14h 08m  | 6h 35m  |

## F    COMPUTE INFRASTRUCTURE

We list our hardware specifications as follows:

- GPU: NVIDIA RTX 4090 ($\times 8$)
- CPU: AMD EPYC 9554

We also list our software specifications as follows:

- Python: 3.8.18
- Pytorch: 1.12.1+cu113
- Numpy: 1.22.4
- Gym: 0.22.0
- MuJoCo: 2.0
- D4RL: 1.1

## G    FINE-TUNING TIME COST OF PES

We demonstrate the efficiency of PES by comparing the average online fine-tuning time cost of the base algorithms including IQL, CQL, TD3-BC w/ and w/o PES on 9 D4RL MuJoCo datasets. The result is presented in Table 7. We can see that after applying PES, the time cost of base algorithms does not increase significantly, which indicates the computational efficiency of PES.

## H    MORE EXPERIMENTAL RESULTS

In this part, we provide more experimental results missing from the main text. In Section H.2, we provide the learning curves for base algorithms w/ and w/o PES on MuJoCo and Antmaze datasets. In Section H.3, we present the parameter study results of PES on wider D4RL datasets. In Section H.4, we vary the design choice for PES, conducting extensive ablation studies on several D4RL datasets. In Section H.5, we use IQL and AWAC as our base algorithms and conduct experiments on challenging Adroit datasets. In Section H.6, we verify the effectiveness of PES to the heterogeneous case where different RL algorithms are applied for offline and online phases. In Section H.7, we provide the full experimental results in previous sections with standard deviations.

### H.1    EVALUATION OF DATA DIVERSITY

To assess the impact of dynamically adjusting the selection threshold $\epsilon$ on data diversity, we perform a comparative experiment on `hopper-medium-v2` dataset. In the control group, we maintain the threshold coefficient $\alpha$ at its default value of 1, allowing the threshold to be adjusted. Conversely, for the experimental group, we set $\alpha$ to 0, thereby fixing the selection threshold $\epsilon$ throughout the online phase. Figure 9 shows the data distribution within the replay buffer for both groups at the end of the online phase, utilizing t-SNE for visualization. It is evident that the group with dynamic threshold adjustment exhibits a more diversed data distribution, suggesting that such adjustments during the online phase enhance data diversity by incorporating a broader range of data qualities.

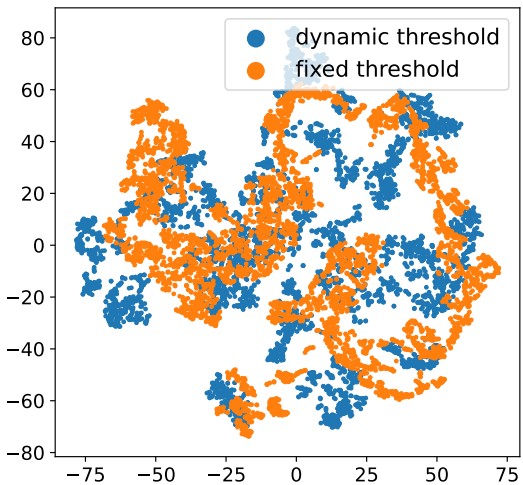

Figure 9: Data distribution comparison within the replay buffer between dynamic threshold and fixed threshold experiment.

## H.2 LEARNING CURVES

We provide the detailed learning curves missing from Section 4.2. Specifically, we supplement the performance comparison between base algorithms (CQL, TD3-BC, Cal-QL, PEX, AWAC) w/ and w/o PES on D4RL MuJoCo and Antmaze datasets. Figure 10, Figure 11, Figure 12, Figure 13, Figure 14 show the experimental results of CQL, TD3-BC, Cal-QL, PEX, and AWAC, respectively.

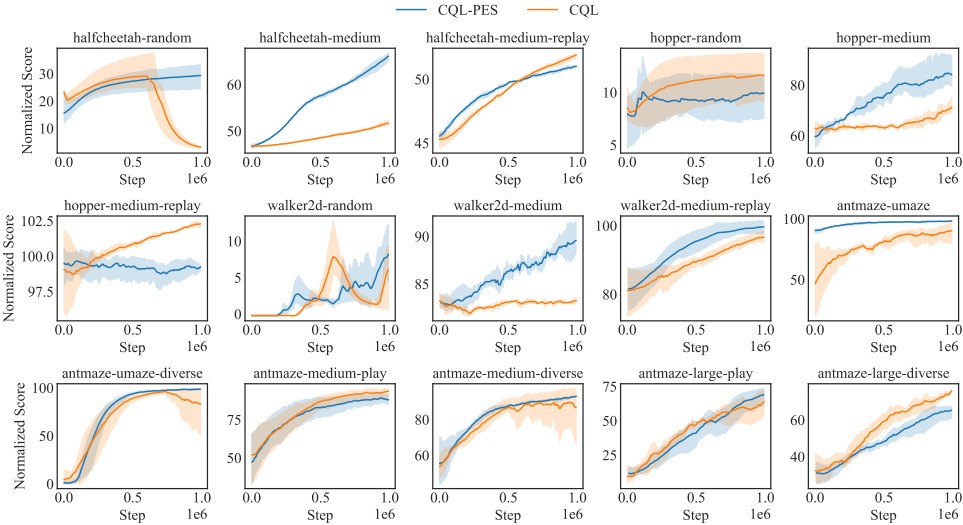

Figure 10: Normalized score comparison for CQL and CQL-PES on 15 datasets of D4RL benchmark. The solid line is the average return, and the shaded area is the 95% confidence interval. The experiments are run with 5 random seeds.

## H.3 WIDER PARAMETER STUDY

In this part, we include additional experimental results of hyperparameter sensitivity in terms of the capacity of the queue $N$, threshold coefficient $\alpha$, number of neighbors $k$, and sampling coefficient

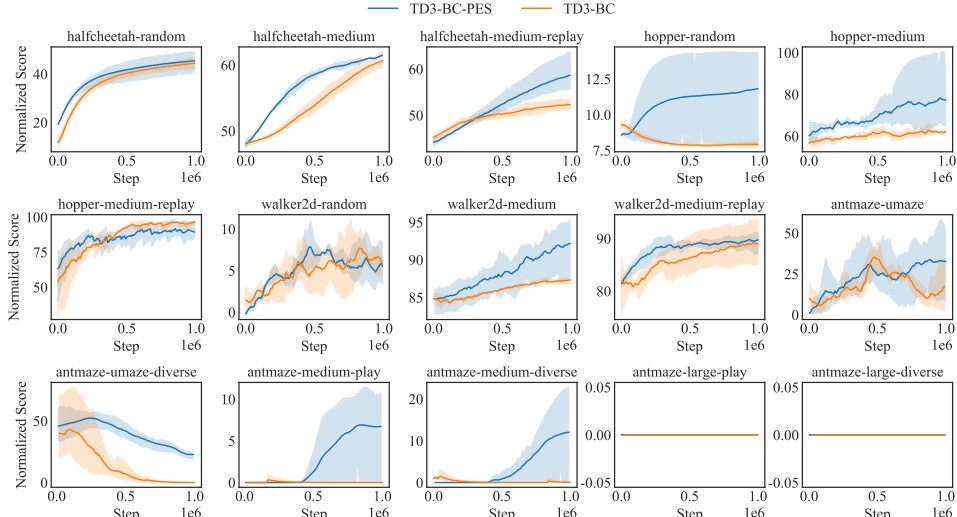

Figure 11: Normalized score comparison for TD3-BC and TD3-BC-PES on 15 datasets of D4RL benchmark. The solid line is the average return, and the shaded area is the 95% confidence interval. The experiments are run with 5 random seeds.

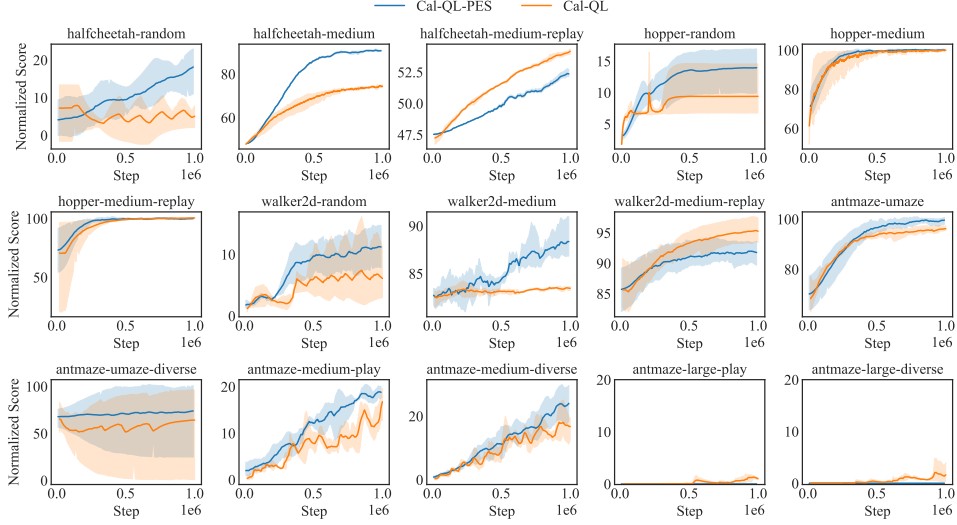

Figure 12: Normalized score comparison for Cal-QL and Cal-QL-PES on 15 datasets of D4RL benchmark. The solid line is the average return, and the shaded area is the 95% confidence interval. The experiments are run with 5 random seeds.

$\eta$, which are missing from the main text due to the space limit. Note that we use IQL as the base algorithm for PES, and the other hyperparameter setting is aligned with Section E.3.

**Capacity of the queue $N$.** $N$ represents the number of trajectories maintained in the priority queue. Too small $N$ can impede the fine-tuning performance as most of the samples may get rejected, while too large $N$ may result in a decrease in trajectory quality (since numerous samples are admitted) and introduce more computational burden (since the search dataset becomes larger). In the main text, we conduct experiments on Antmaze domain and find that $N = 10$ is a proper value. We conduct additional two experiments on MuJoCo datasets, `hopper-medium-v2` and `walker2d-medium-v2`. We vary $N$ across $\{1, 10, 100\}$ and present the results in Figure 15(a).

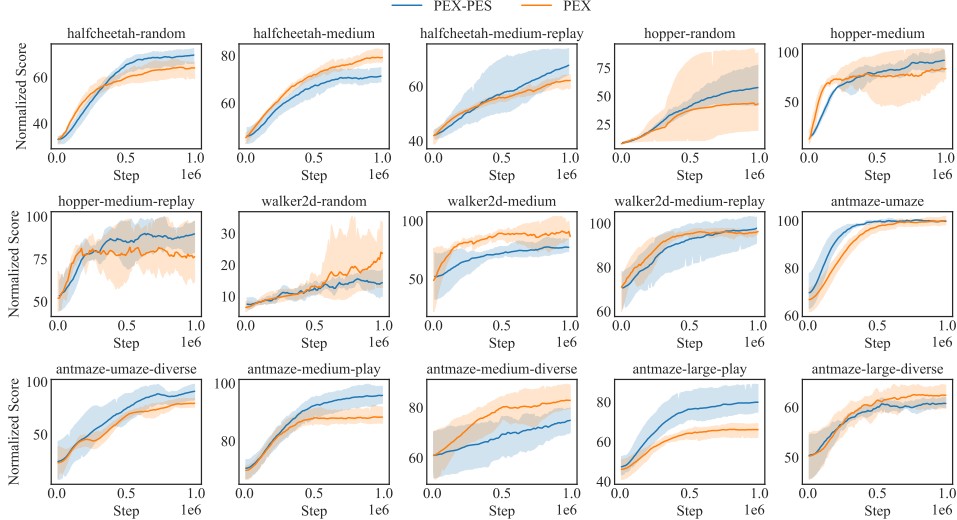

Figure 13: Normalized score comparison for PEX and PEX-PES on 15 datasets of D4RL benchmark. The solid line is the average return, and the shaded area is the 95% confidence interval. The experiments are run with 5 random seeds.

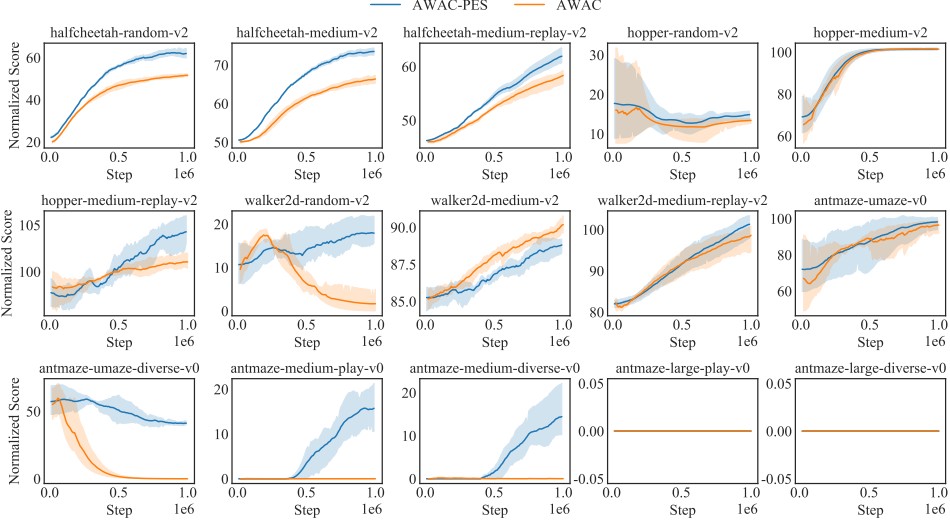

Figure 14: Normalized score comparison for AWAC and AWAC-PES on 15 datasets of D4RL benchmark. The solid line is the average return, and the shaded area is the 95% confidence interval. The experiments are run with 5 random seeds.

The results show that a small $N$, i.e., $N = 1$ or a large $N$, i.e, $N = 100$, can not lead to a performance improvement as significant as $N = 10$. Therefore, we simply set $N = 10$ in our experiments.

**Threshold coefficient $\alpha$.** $\alpha$ controls the threshold of sample selection. A too small $\alpha$ can lead to an overly strict sample selection, e.g., filtering out most of the samples, while a too large $\alpha$ can render the sample selection ineffective, e.g., admitting too many online samples. We vary $\alpha$ across $\{0.1, 1, 10\}$ and conduct additional experiments on hopper-medium-v2 and walker2d-medium-v2 datasets. The experimental results are presented in Figure 15(b). We find that the impact of $\alpha$ depends on specific datasets. For walker2d-medium-v2, three different values of $\alpha$ do not introduce significant differences in final performance. However, we can

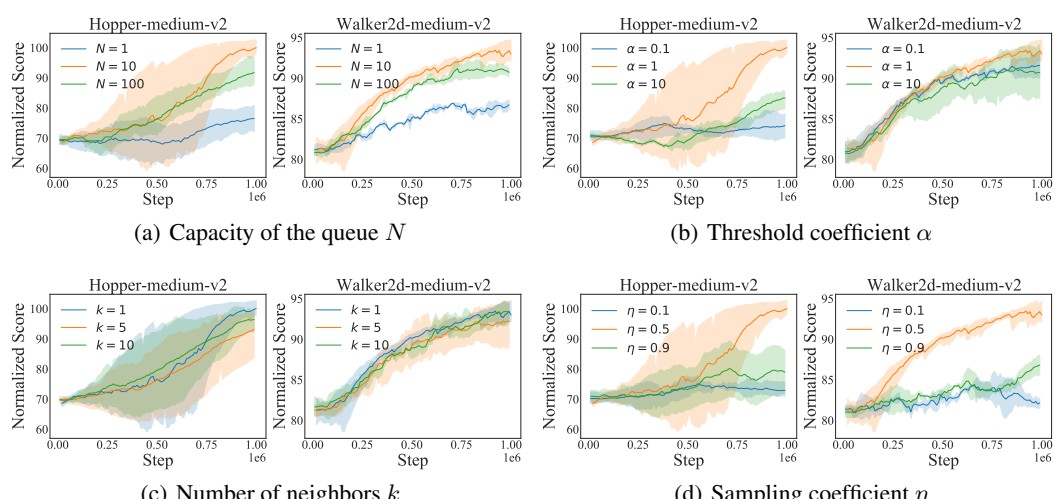

Figure 15: Parameter study results of $N$, $\alpha$, $k$, $\eta$ on wider datasets.

observe that setting $\alpha = 1$ can achieve good performance on these datasets. So we can simply set $\alpha = 1$.

**Number of neighbors $k$.** $k$ is the hyperparameter introduced in $k$-nearest neighbor algorithms. We observe PES is robust to the value of $k$ in Section 4.5. To investigate whether this conclusion holds for wider range of datasets, we conduct additional experiments on `hopper-medium-v2` and `walker2d-medium-v2` datasets, and the results in Figure 15(c) show that the value of $k$ has minor influence on the performance. We simple set $k = 1$ for all of our experiments.

**Sampling coefficient $\eta$.** $\eta$ controls the proportion of offline and online samples used for fine-tuning. A larger $\eta$ implies using a greater proportion of offline samples. If $\eta$ is too small, training instability may occur due to distribution shift. If $\eta$ is too large, the performance improvement may be slow. We vary $\eta$ across $\{0.1, 0.5, 0.9\}$ and conduct experiments on `hopper-medium-v2` and `walker2d-medium-v2` datasets. The experimental results are shown in Figure 5(d). We observe that $\eta = 0.5$ can achieve relatively good performance, while too small or too large $\eta$ both result in decreased sample efficiency.

## H.4 ABLATION STUDY

In the main text, we examine the significance of `Return-Prioritized Selection` and `Priority Queue Update`. In this part, we mainly examine two other design choices in PES: `Search Vector` and `Distance Measure`.

**Search Vector.** This determines the search space of $k$-nearest neighbor search for PES. For the default setting in our experiments, we search in the state-action space, yielding the search vector of $(s \oplus a)$. In addition to $(s \oplus a)$, one can also choose other search vectors, such as $(s \oplus a \oplus s')$ and $(s \oplus s')$. To examine whether different search vectors matter for PES, we change the choice of search vector and conduct extensive experiments on several D4RL datasets. We present the results in Table 8, and the results show that the impact of different search vector seems to depend on specific datasets. Some datasets (e.g., `hopper-medium-v2`, `antmaze-umaze-diverse-v0`) prefer the choice of $(s \oplus a)$, while it is better to use $(s \oplus a \oplus s')$ as the search vector for tasks like `halfcheetah-medium-v2`, `walker2d-medium-v2`, and `antmaze-large-diverse-v0`. However, considering both performance and computational burden, we simply use $(s \oplus a)$ for all the experiments.

**Distance Measure.** The default distance measure used in PES is `Euclidean distance`. One can certainly utilize other common distance measures like `Manhattan distance`, `Chebyshev distance`, etc. To examine the impact of different distance measures on PES, we replace the default `Euclidean distance` with `Manhattan distance` and `Chebyshev distance`,

Table 8: Performance comparison for IQL-PES with different search vectors on various D4RL datasets. All the experiments are run with 5 random seeds, and the superior scores are in bold and highlighted in green.

| Task Name | $s \oplus a$ | $s \oplus s'$ | $s \oplus a \oplus s'$ |
|---|---|---|---|
| halfcheetah-medium-v2 | 68.8±3.3 | 70.7±1.6 | **71.2±1.1** |
| hopper-medium-v2 | **100.0±1.1** | 94.2±2.4 | 91.0±1.9 |
| walker2d-medium-v2 | 93.6±1.3 | 84.1±2.9 | **95.7±0.3** |
| antmaze-umaze-diverse-v0 | **81.0±17.2** | 74.3±19.4 | 77.4±14.2 |
| antmaze-medium-diverse-v0 | **88.4±5.6** | 79.6±3.3 | 84.2±7.1 |
| antmaze-large-diverse-v0 | 66.8±6.1 | 62.1±4.4 | **73.2±7.0** |

Table 9: Performance comparison for IQL-PES with different distance measures on various D4RL datasets. All the experiments are run with 5 random seeds, and the superior scores are in bold and highlighted in green.

| Task Name | Euclidean | Manhattan | Chebyshev |
|---|---|---|---|
| halfcheetah-medium-v2 | **68.8±3.3** | 60.4±4.0 | 67.0±2.7 |
| hopper-medium-v2 | **100.0±1.1** | 98.4±2.3 | 92.1±1.6 |
| walker2d-medium-v2 | 93.6±1.3 | **95.2±2.1** | 88.1±2.4 |
| antmaze-umaze-diverse-v0 | **81.0±17.2** | 77.1±21.3 | 73.2±13.2 |
| antmaze-medium-diverse-v0 | 88.4±5.6 | 84.3±3.1 | **91.3±4.9** |
| antmaze-large-diverse-v0 | **66.8±6.1** | 61.2±7.7 | 64.3±4.5 |

which is easily done by changing the `metric` parameter of `KDTree`. We conduct extensive experiments on D4RL datasets and the experimental results are presented in Table 9. It is observed that the default `Euclidean distance` can bring good performance, therefore we use `Euclidean distance` for our experiments.

## H.5 EXPERIMENTAL RESULTS ON ADROIT DATASETS

In this part, we present missing experimental results for PES on D4RL Adroit datasets.

**Experimental Setup.** The base algorithms we use are IQL and AWAC. We evaluate the base algorithms w/ and w/o combined with PES on all 12 D4RL Adroit datasets introduced in Section D.3. The offline gradient steps and online environmental steps are both set to be 1M. The other hyperparameter setup is listed in Table 6.

**Experimental Results.** We present the experimental results in Figure 16 and Figure 17. Figure 16 depicts the performance comparison between IQL and IQL-PES, and Figure 17 illustrates the performance comparison between AWAC and AWAC-PES. We can find that PES can benefit IQL and AWAC on most of 12 Adroit datasets, clearly verifying the effectiveness and advantages of PES on challenging Adroit datasets.

## H.6 HETEROGENEOUS OFFLINE-TO-ONLINE EXPERIMENTS WITH PES

We have shown the effectiveness of PES to the case where offline and online algorithms are the same in Section 4.1 and Section 4.2. In this part, we further investigate whether PES can benefit heterogeneous RL algorithms, i.e., different RL algorithms are used for offline and online phases. For example, we can remove the behavior cloning term from TD3-BC or remove the conservatism term from CQL during the online phase, giving rise to H-TD3-BC and H-CQL. Then, we can integrate PES into H-TD3-BC and H-CQL to examine the effectiveness of PES to this heterogeneous case.

**Experimental Setup.** We use H-TD3-BC and H-CQL as the base algorithms and evaluate them w/ and w/o combined with PES on 9 D4RL MuJoCo datasets. We keep the original hyperparameters unchanged and the only difference is the removal of the behavior cloning term and conservatism term during the online phase.

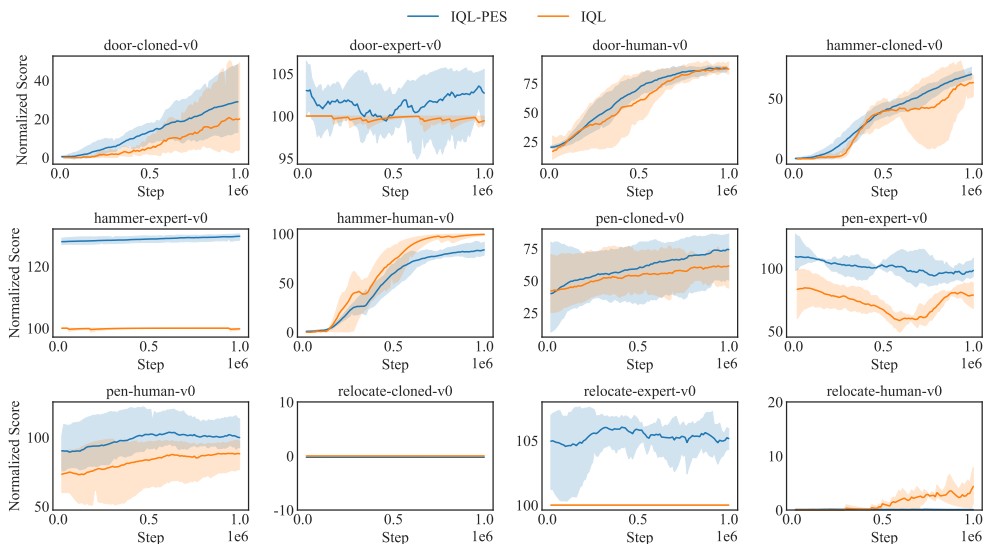

Figure 16: Normalized score comparison for IQL and IQL-PES on 12 D4RL Adroit datasets. The solid line is the average return, and the shaded area is the 95% confidence interval. The experiments are run with 5 random seeds.

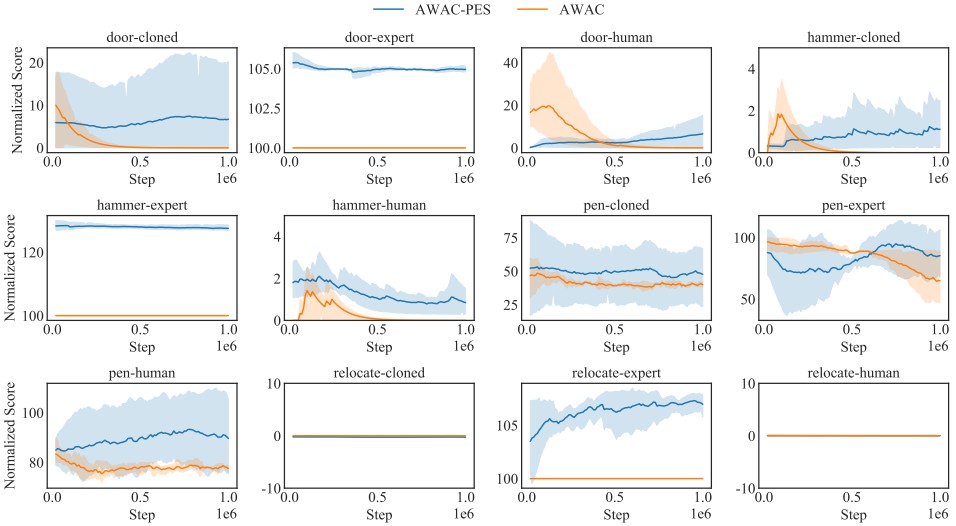

Figure 17: Normalized score comparison for AWAC and AWAC-PES on 12 D4RL Adroit datasets. The solid line is the average return, and the shaded area is the 95% confidence interval. The experiments are run with 5 random seeds.

**Experimental Results.** We present the experimental results in Figure 18 and Figure 19. We can see that due to the heterogeneity of the algorithm form during online phase, the performance of the original algorithm may collapse. This happens to both H-CQL and H-TD3-BC on `hopper-medium-v2` dataset. After incorporating PES, the performance collapse is mitigated, and more significant performance improvements are observed in most of 9 MuJoCo datasets. This indicates PES is also effective in case where different RL algorithms are employed in offline and online stages.

## H.7 FULL EXPERIMENTAL RESULTS WITH STANDARD DEVIATION

In this part, we supplement the full experimental results with standard deviation from Section 4.1, Section 4.2 and Section H.5. The results are presented in Table 10.

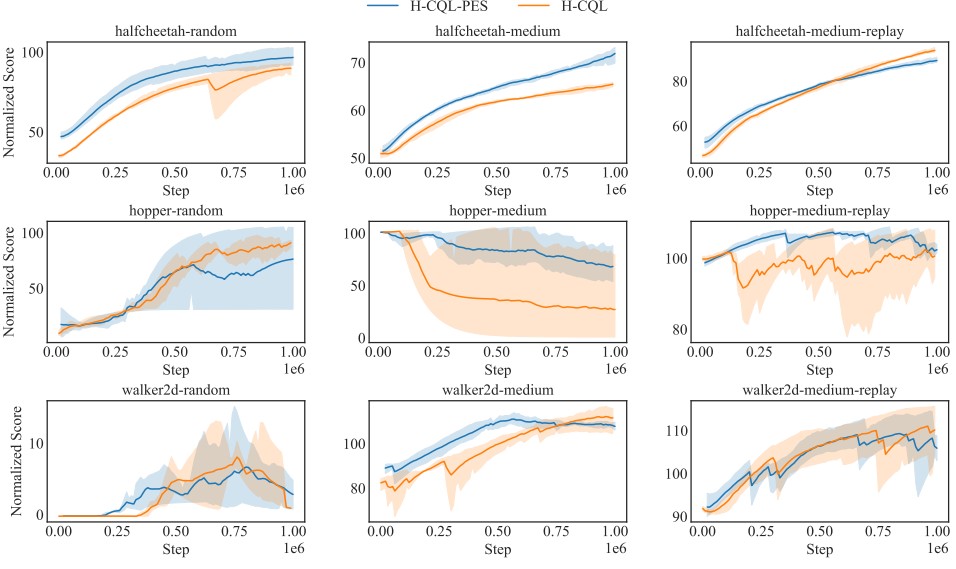

Figure 18: Normalized score comparison for H-CQL and H-CQL-PES on 9 D4RL MuJoCo datasets. The solid line is the average return, and the shaded area is the 95% confidence interval. The experiments are run with 5 random seeds.

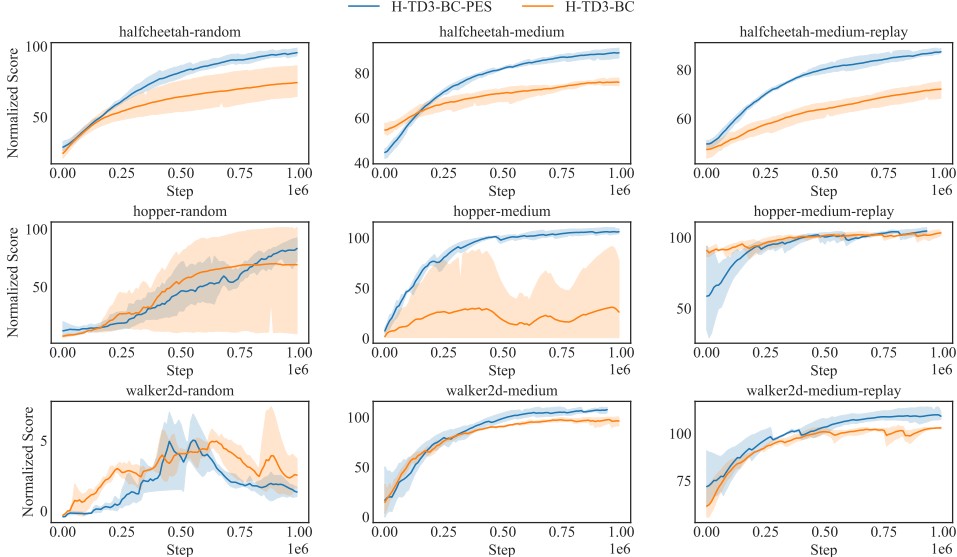

Figure 19: Normalized score comparison for H-TD3-BC and H-TD3-BC-PES on 9 D4RL MuJoCo datasets. The solid line is the average return, and the shaded area is the 95% confidence interval. The experiments are run with 5 random seeds.

Table 10: Performance comparison for base algorithms w/ (denoted as "Ours") and w/o (denoted as "Base") PES with standard deviation on D4RL benchmark. We abbreviate "halfcheetah" as "half", "random" as "r", "medium" as "m", "medium-replay" as "m-r". The version of locomotion datasets is "-v2", and the version of Antmaze datasets is "-v0". We report the normalized score for each dataset. All the experiments are run with 5 random seeds, and the superior normalized scores are in bold and highlighted in green.

| Task Name | AWAC Base | AWAC Ours | PEX Base | PEX Ours | Cal-QL Base | Cal-QL Ours | TD3-BC Base | TD3-BC Ours | CQL Base | CQL Ours | IQL Base | IQL Ours |
|---|---|---|---|---|---|---|---|---|---|---|---|---|
| half-r | 52.4±0.8 | **61.1±1.8** | 64.2±2.1 | **69.6±1.7** | 3.2±1.9 | **18.2±2.3** | 44.3±2.4 | **45.1±4.7** | 0.0±0.0 | **30.0±4.4** | 51.4±0.3 | **54.2±0.2** |
| half-m | 67.2±1.5 | **73.5±1.3** | **79.0±1.4** | 72.1±1.5 | 73.1±0.1 | **90.5±0.1** | 61.5±1.7 | **63.4±0.6** | 52.5±1.1 | **64.7±1.2** | 57.4±0.1 | **68.8±3.3** |
| half-m-r | 59.2±3.8 | **62.3±3.6** | 62.5±1.1 | **68.3±3.2** | **54.7±0.2** | 52.2±0.3 | 52.3±1.6 | **58.7±3.3** | **53.6±0.4** | 52.1±0.5 | 51.0±0.2 | **57.7±0.4** |
| hopper-r | 13.2±0.7 | **14.8±0.9** | 41.2±28.5 | **58.4±14.2** | 9.6±4.8 | **14.4±4.5** | 7.7±0.2 | **12.2±4.8** | **11.7±2.4** | 10.0±1.4 | **20.7±4.4** | 18.6±2.1 |
| hopper-m | 101.0±0.01 | 101.0±0.01 | 83.1±12.9 | **91.2±8.2** | 100.0±0.0 | 100.0±0.0 | 62.1±1.2 | **79.3±19.5** | 72.1±2.6 | **81.4±14.3** | 76.1±4.1 | **100.0±1.1** |
| hopper-m-r | 101.3±2.4 | **104.5±3.9** | 77.2±14.8 | **90.0±7.1** | 100.0±0.0 | 100.0±0.0 | **93.1±1.8** | 87.6±11.3 | **102.4±1.1** | 99.1±2.7 | 101.0±0.1 | **102.3±1.3** |
| walker2d-r | 2.4±1.6 | **18.6±3.1** | **24.1±7.8** | 14.7±3.4 | 6.4±5.1 | **11.3±3.7** | 5.4±2.1 | 5.4±2.7 | 6.6±0.8 | **8.4±1.7** | **9.4±0.1** | 7.6±2.2 |
| walker2d-m | **90.1±0.8** | 88.9±1.2 | **86.4±9.5** | 77.3±4.8 | 83.5±0.3 | **88.2±2.3** | 87.5±0.4 | **92.1±4.8** | 83.2±0.2 | **89.6±3.7** | 87.7±2.4 | **93.6±1.3** |
| walker2d-m-r | 98.5±3.1 | **101.3±2.0** | 94.3±0.4 | **98.1±2.9** | **95.1±2.7** | 91.9±2.1 | 88.3±4.2 | **90.2±1.5** | 97.6±0.3 | **99.8±0.7** | **103.4±2.1** | 99.7±0.5 |
| umaze | 97.3±1.4 | **99.7±0.8** | 100.0±1.4 | 100.0±0.3 | 95.9±1.4 | **99.8±0.4** | 17.4±7.7 | **33.4±15.4** | 90.8±4.9 | **99.5±0.2** | 96.4±0.1 | **97.2±0.1** |
| umaze-diverse | 0.0±0.0 | **42.6±0.2** | 79.6±4.6 | **91.7±7.1** | 64.2±48.1 | **72.3±33.5** | 0.0±0.0 | **23.7±1.2** | 77.2±22.1 | **100.0±0.0** | 48.2±19.5 | **81.0±17.2** |
| medium-diverse | 0.0±0.0 | **13.8±6.6** | **83.0±8.6** | 75.1±7.1 | 16.8±4.7 | **24.3±3.6** | 0.0±0.0 | **12.1±7.1** | 87.6±16.7 | **93.2±1.1** | **91.7±0.7** | 88.4±5.6 |
| medium-play | 0.0±0.0 | **15.6±5.2** | 88.1±5.5 | **95.3±4.9** | 17.2±2.8 | **19.0±0.8** | 0.0±0.0 | **7.4±3.0** | **93.1±2.3** | 88.1±3.5 | 90.1±0.3 | **92.3±0.2** |
| large-diverse | 0.0±0.0 | 0.0±0.0 | **63.4±2.1** | 61.0±0.8 | **1.5±0.2** | 0.0±0.0 | 0.0±0.0 | 0.0±0.0 | **76.1±1.0** | 66.3±2.9 | 62.3±5.3 | **66.8±6.1** |
| large-play | 0.0±0.0 | 0.0±0.0 | 67.2±3.4 | **80.1±11.4** | 1.1±1.0 | 0.0±0.0 | 0.0±0.0 | 0.0±0.0 | 63.3±8.9 | **69.2±7.1** | 59.2±2.4 | **63.5±4.7** |
| door-cloned | 0.0±0.0 | **8.6±7.7** | – | – | – | – | – | – | – | – | 18.1±13.5 | **24.6±9.8** |
| door-expert | 100.0±0.0 | **104.8±0.2** | – | – | – | – | – | – | – | – | 99.6±0.2 | **103.4±1.7** |
| door-human | 0.0±0.0 | **10.3±5.5** | – | – | – | – | – | – | – | – | 82.1±2.6 | **82.4±1.1** |
| hammer-cloned | 0.0±0.0 | **1.2±0.8** | – | – | – | – | – | – | – | – | 72.1±3.6 | **75.4±1.3** |
| hammer-expert | 100.0±0.0 | **124±1.0** | – | – | – | – | – | – | – | – | 100.0±0.0 | **126±0.5** |
| hammer-human | 0.0±0.0 | **1.4±0.3** | – | – | – | – | – | – | – | – | **99.2±0.2** | 87.4±2.1 |
| pen-cloned | 44.8±0.5 | **49.1±18.5** | – | – | – | – | – | – | – | – | 59.7±6.3 | **73.1±8.2** |
| pen-expert | 61.8±14.2 | **82.2±18.2** | – | – | – | – | – | – | – | – | 73.5±5.2 | **100.4±3.2** |
| pen-human | 78.7±0.7 | **90.5±8.1** | – | – | – | – | – | – | – | – | 83.3±6.2 | **101.2±2.8** |
| relocate-cloned | 0.0±0.0 | 0.0±0.0 | – | – | – | – | – | – | – | – | 0.0±0.0 | 0.0±0.0 |
| relocate-expert | 100.0±0.0 | **107.3±1.4** | – | – | – | – | – | – | – | – | 100.0±0.0 | **105.2±2.0** |
| relocate-human | 0.0±0.0 | 0.0±0.0 | – | – | – | – | – | – | – | – | **5.9±1.1** | 0.0±0.0 |

