# OpenReview forum: "Offline-to-Online Reinforcement Learning with Prioritized Experience Selection"
_ICLR.cc/2025/Conference — Submitted to ICLR 2025_

### Official Review · Reviewer_4p5p · 2024-10-30

**Soundness:** 2
**Presentation:** 2
**Contribution:** 2
**Rating:** 5
**Confidence:** 4

**Summary:**

This paper proposed the Prioritized Experience Selection (PES) for offline-to-online RL. During the online stage, PES maintains a priority queue for high-return trajectories, and only selects online samples that are close to the samples in the queue for fine-tuning. PES shows strong performances in experiments on the D4RL benchmark.

**Strengths:**

- The writing is clear and the paper is easy to follow.
- The proposed method is simple and can be combined with different algorithms.
- PES shows promising results in the experiments.

**Weaknesses:**

- Background Section is too simple.
- Line 34: "learns the optimal policy from a previously collected dataset" is incorrect. We cannot learn an optimal policy from a low-quality offline dataset.
- Missing recent baselines: BOORL (https://github.com/YiqinYang/BOORL), RLPD (https://github.com/ikostrikov/rlpd), SO2 (https://github.com/opendilab/SO2), FamO2O (https://github.com/LeapLabTHU/FamO2O).
- PES needs to search a k-nearest neighbours in the queue when we have a new online transition, which increases the computational cost. Further, it also introduces extra hyper-parameters, i.e., the size of the queue and the neighbour size k.
- Insufficient details are provided for readers to understand the specifics of the two baselines in Section 3.1.
- Theorem 3.1 is less practical. Firstly, the difference of empirical MDP for two consecutive steps are usually negligible in practice. Further, a more close related theory analysis should focused on the distribution shift between the online policy and the data in the replay buffer. How does distribution shift influence the value & policy learning is not analyzed.
- Line 185: "and the sample efficiency can be boosted (since only high-quality samples are selected)". The description is misleading. We use less samples during the training, but we still have the unused samples during the rollout. From the perspective of environmental steps, we still have the same number of total samples.
- Line 186: "the computational load is also significantly reduced ..." is also inaccurate. PES mainly saves memory usage, but the computational cost (FLOPs) is not reduced.
- The effectiveness of PES is not stable. Sometimes adding PES make the performance worse and sometimes PES can not improve the performance.

**Questions:**

- In the abstract, "In this way, the distribution shift issue can be mitigated...". The queue only contains high-return trajectories, which corresponds to a narrower state distribution. The distribution shift issue is mitigated only when the online policy can generate high-return trajectories as in the queue. However, during the offline-to-online fine-tuning the online policy is dynamic and it is not always a high-performing one. I think the distribution shift issue is not guaranteed to be mitigated.
- Line 67: "we make sure that the samples used for fine-tuning stay close to the previously encoutered samples". We can also achieve this by using a smaller FIFO buffer to only store online samples. So there is a missing baseline: fine-tune the agent with a smaller FIFO buffer that only stores online samples.
- What's the detail of IQL-Expert baseline in Section 3.1? I did a quick experiment to fine-tune an IQL agent, which is first pretrained on the halfcheetah-medium-v2 dataset, with the halfcheetah-expert-v2 dataset for 1e6 steps. The curve is very different from IQL-Expert in Figure 1(right). In my experiment, the normalized score increases from 47 to 94. Also, there is a severe distribution shift issue in my experiment. However, this does not prevent the agent from improving its performance from 47 to 94. Therefore, the main point of this work on the problem of distribution shift is not complete. The claim in Section 3.1 is not correct as well. Distribution shift is not always harmful. We can still train a high-performing policy even with the distribution shift issue. A deeper discussion about when it hurts and when it doesn't would increase this paper's contribution.
- If we use pixel-based inputs, will the computational cost to find k-nearest neighbors be large? How does the proposed method perform with pixel-based inputs, i.e., final performance and computational cost?
- Line 71: "we adapt the selection threshold throughout the online phase to ensure data diversity". How can we prove this point? Any experiments to support this claim?

---

> ### Author Response · Authors · 2024-11-23
> **Response to Reviewer 4p5p (Part 1)**
>
> We thank the reviewer for insightful comments. Please find our clarifications to the concerns below. If we are able to address the concerns, we hope the reviewer can reconsider the score.
>
> **Concern 1: On the mitigation of distribution shift issue**
>
> First, we want to say that generating high-performance trajectories with online policy is not difficult because the initial trajectories in the priority queue come from the dataset, and the policy has already been pre-trained on the offline dataset, usually performing better than the behavioral policy of the offline dataset. Therefore, the priority queue is likely to be continuously updated. Additionally, we dynamically relax the selection threshold, which means that more diverse data will be added to the replay buffer, enhancing the exploration capability of the policy.
>
> **Concern 2: Baseline of FIFO buffer**
>
> We thank the suggestion of  the additional baseline. We implement the FIFO baseline as follows: instead of initializing the priority queue with the offline dataset, we continuously added online sampled trajectories in a FIFO manner, with the rest of the process being the same as PES. We conduct experiments on 12 D4RL MuJoCo datasets and present the results below. It can be seen that PES significantly outperforms the FIFO baseline, achieving higher performance on 9 out of 12 datasets. We believe this is because, although FIFO mitigates distribution shift, it cannot ensure the high quality of the samples used for training, as online samples may be of low quality. In contrast, PES can both mitigate distribution shift and ensure the high quality of samples.
>
> | dataset | IQL+PES | FIFO |
> | :--------  | :-----  | :----:  |
> | half-r |54.2 $\pm$ 0.2 |**55.6 $\pm$ 0.3**|
> | half-m | **68.8 $\pm$ 3.3** | 57.4 $\pm$ 3.7 |
> | half-mr |**57.7 $\pm$ 0.4** | 52.9 $\pm$ 0.5|
> | half-me | **73.4 $\pm$ 0.2** | 66.8 $\pm$ 0.3 |
> | hopper-r | **18.6 $\pm$ 2.1** | 15.4 $\pm$ 2.1 |
> | hopper-m | **100.0 $\pm$ 1.1** | 88.3 $\pm$ 1.9 |
> | hopper-mr | 102.3 $\pm$ 1.3 | **107.2 $\pm$ 1.1** |
> | hopper-me | **107.1 $\pm$ 2.4** | 100.3 $\pm$ 2.2 |
> | walker-r | 7.6 $\pm$ 2.2 | **12.8 $\pm$ 1.0** |
> | walker-m | **93.6 $\pm$ 1.3** | 88.9 $\pm$ 1.1 |
> | walker-mr | **99.7 $\pm$ 0.5** | 94.3 $\pm$ 0.6 |
> | walker-me |**104.2 $\pm$ 0.7** | 97.3 $\pm$ 0.6 |
>
> **Concern 3: The detail of IQL-Expert**
>
> First, we will elaborate on the process of the IQL-Expert baseline. Initially, we load the data from the halfcheetah-medium-v2 dataset into the replay buffer and train the IQL algorithm for 1 million steps, which is the same as the training process for general offline algorithms. Subsequently, we **sequentially** add the data from the halfcheetah-expert-v2 dataset (approximately 1 million data points) to the replay buffer (which still contains medium quality data), and we train the IQL algorithm once for each added data point, totaling 1 million steps. We believe that the reason the reviewer's results differ from ours is that the reviewer fine-tuned directly on the entire dataset of halfcheetah-expert-v2, whereas we sequentially added data to the replay buffer (which is more in line with the situation during online fine-tuning). Secondly, we argue that even if the final performance of the algorithm is high, the initial performance collapse caused by distribution shift is still important. For example, in safe reinforcement learning, we not only need good final performance but also need to satisfy certain safety constraints. Sudden performance collapse may violate safety constraints and bring hidden dangers.
>
> **Concern 4: On the Pixel inputs**
>
> Although PES is not an algorithm designed for pixel inputs, we understand the reviewer's concern. Therefore, we select several datasets from V-D4RL and compare the performance and computation time of IQL and IQL+PES. We present the performance comparison below, which shows that under visual inputs, IQL+PES still outperforms IQL on 3 out of 4 datasets, demonstrating the superiority of PES. As for computation time, the average training time for IQL across the 4 datasets is 9 hours and 14 minutes, while the average training time for IQL+PES is 13 hours and 22 minutes, which is significantly longer than IQL. We acknowledge that the training efficiency under visual inputs is reduced due to KNN search, but this can be mitigated by using visual encoders to reduce dimensionality, which we will consider as future work.
>
> | dataset | IQL | IQL+PES |
> | :--------  | :-----  | :----:  |
> | walker-walk-random |14.2 $\pm$ 4.4 | **19.1 $\pm$ 3.5**|
> | walker-walk-medium | **48.8 $\pm$ 6.3** | 41.4 $\pm$ 2.7 |
> | cheetah-run-random | 17.2 $\pm$ 7.4 | **21.9 $\pm$ 4.5**|
> | cheetah-run-medium | 53.4 $\pm$ 5.2 | **61.8 $\pm$ 7.3** |

---

> > ### Author Response · Authors · 2024-11-23
> > **Response to Reviewer 4p5p (Part 2)**
> >
> > **Concern 5: Experiments of data diversity**
> >
> > We have conducted an additional experiment to evaluate the data diversity with and without dynamically adjusting the selection threshold. We set the threshold coefficient $\alpha$ to 0 and 1 respectively and compare the data distribution within the replay buffer, which can reflect data diversity. We visualize the data distribution via t-SNE and find that dynamically adjusting the threshold can increase the data diversity. For more details, please refer to Appendix H.1 in our revision.

---

> ### Comment · Reviewer_4p5p · 2024-11-26
>
> Many thanks to the authors for the explanations and additional experiments. I would like to raise my score to 5. I didn't raise to a higher score because: (1) I still find the Theorem 3.1 is less directly connected to the central claim of this paper. I would like to see some analysis about how does the current policy distribution shift affect the policy learning, and when does it hurt. (2) Authors mentioned about the initial performance collapse issue in offline-to-online fine-tuning in the rebuttal, but this point is not well-discussed in the main paper. A closely related work on this issue is not compared (https://github.com/YiqinYang/BOORL). (3) Adding one sample at a time is closer to the fine-tuning setting, however, adding a sample from the expert dataset is not close to the fine-tuning setting. If we use the offline policy to collect online samples, we can only obtain some similar samples from the offline dataset, which has a much smaller distribution shift than the one from the expert dataset.

---

> ### Author Response · Authors · 2024-11-28
> **Response to Reviewer 4p5p**
>
> We thank the reviewer for raising the score, and we are pleased to address the remaining concerns. Please find our clarifications to the concerns.
>
> **Concern 1: Theorem 3.1 is less connected to the central claim**
>
> Our central claim is that distribution shift may lead to performance collapse, and we introduce PES to mitigate this issue. Theorem 3.1 is connected to our central claim as it reveals that when the empirical MDP between two consecutive steps differs greatly (i.e., the distribution shift is severe), performance collapse may occur. From theorem 3.1, we can also infer that if the distribution shift between two steps is severe and the empirical distribution is greatly different from the true distribution, the policy may not improve steadily, thus hindering policy learning. We note that theoretically analyzing the impact of policy distribution shift is challenging since the policy is optimized via SGD. If we were to analyze the deterministic policy improvement, many assumptions would need to be introduced, and these assumptions may not hold under practical scenarios. Therefore, the theoretical guarantee of policy improvement is beyond the scope of this paper, but we believe this is an interesting point and can be further investigated in our future work.
>
> **Concern 2: The initial performance collapse is not well-discussed**
>
> Firstly, we respectfully argue that we have discussed the phenomenon of initial performance collapse in our paper since it is our central claim. We mention this issue in our paper in line 13, 45, 310 and so on, and our theoretical analysis is also related to the initial performance collapse phenomenon. Secondly, we thank the reviewer for providing BOORL as another baseline. We therefore integrate PES into BOORL and conduct experiments on various D4RL datasets, and present the results below. We can see that PES can benefit BOORL in 8 out of 12 datasets, indicating the effectiveness and versatility of PES. Although the improvements are not quite significant on some datasets (e.g., from 97.2 to 99.5 on half-me), this is because BOORL is already well-performing on D4RL datasets.
>
> | dataset | BOORL+PES | BOORL |
> | :--------  | :-----  | :----:  |
> | half-r |91.2 $\pm$ 0.9 |**94.1 $\pm$ 1.5**|
> | half-m | **103.2 $\pm$ 2.3** | 96.6 $\pm$ 3.7 |
> | half-mr |89.7 $\pm$ 0.4 | **93.2 $\pm$ 0.5**|
> | half-me | **99.5 $\pm$ 2.2** | 97.2 $\pm$ 2.3 |
> | hopper-r | **85.6 $\pm$ 2.0** | 76.3 $\pm$ 1.2 |
> | hopper-m | 106.0 $\pm$ 1.1 | **110.3 $\pm$ 0.9** |
> | hopper-mr | **114.3 $\pm$ 1.3** | 111.5 $\pm$ 2.1 |
> | hopper-me | **109.1 $\pm$ 2.0** | 102.3 $\pm$ 2.2 |
> | walker-r | **92.6 $\pm$ 3.2** | 87.4 $\pm$ 4.2 |
> | walker-m | 103.6 $\pm$ 2.3 | **109.8 $\pm$ 1.2** |
> | walker-mr | **117.7 $\pm$ 0.5** | 113.3 $\pm$ 0.6 |
> | walker-me |**116.2 $\pm$ 0.7** | 114.2 $\pm$ 0.6 |
>
> **Concern 3: Adding samples from expert dataset is not close to fine-tuning**
>
> Yes, we agree that adding samples from expert dataset is not close to fine-tuning manner. However, the aim of the motivating example is not to simulate the fine-tuning phase, but to create a setting with severe distribution shift to better illustrate the superiority of PES, although this setting may not occur in practice. We add samples from expert dataset since the samples in the expert dataset has a distinct distribution shift from the medium dataset.

---

### Official Review · Reviewer_jBYT · 2024-11-02

**Soundness:** 2
**Presentation:** 3
**Contribution:** 2
**Rating:** 5
**Confidence:** 4

**Summary:**

This paper introduces Prioritized Experience Selection (PES) to address the distribution shift challenge in offline-to-online reinforcement learning (O2O RL). PES selectively uses high-return samples from online interactions to fine-tune the policy, mitigating performance degradation and enhancing learning efficiency. Unlike prior methods that rely on complex models or ensembles, PES is lightweight and adaptable to various algorithms. Experiments demonstrate that PES consistently outperforms traditional approaches across multiple O2O RL benchmarks, achieving stable and improved performance.

**Strengths:**

- By selectively using high-return samples, PES helps prevent performance degradation (unlearning) and enhances overall learning stability.
- PES doesn’t require complex models or ensemble methods, making it lightweight and computationally efficient compared to other approaches.
- PES is adaptable and can be easily integrated into various offline-to-online reinforcement learning algorithms, improving their effectiveness without altering core structures.
- By focusing on the most relevant online samples, PES allows policies to quickly adapt to new data and dynamic environments.

**Weaknesses:**

- The sample selection criteria for PES appear somewhat naive, based solely on the distance (difference) between samples in the priority queue. It seems unlikely that a distance-based comparison can consistently guarantee the selection of good samples. Potentially promising new samples might be filtered out due to this distance criterion, which could lead to missed learning opportunities through exploration, ultimately limiting long-term performance.
- The base score of Cal-QL [1] appears unusual. When examining the Cal-QL graph from the original Cal-QL paper, both CQL and Cal-QL achieve scores above 80 in the Antmaze Large environment after 1M finetuning, which aligns with my experience running the official Cal-QL code. Could there be a reason why the Cal-QL score for Antmaze is so low?
- It’s promising that this algorithm performs well even without using an ensemble. However, I wonder if performance could be significantly enhanced by substituting only the balanced replay with PES in configurations that already use both an ensemble and balanced replay. With advancements like JAX accelerating RL training, computational demands might reduce over time. Although achieving strong performance without an ensemble is beneficial, assessing whether a small ensemble combined with the proposed method could improve overall results would ultimately be valuable for the community. From this standpoint, it’s worthwhile to consider integrating with offline-to-online algorithms like Off2on [2], which, despite some computational demands, achieve high performance.
- Furthermore, studies like ReBRAC [3] demonstrate that integrating multiple techniques can yield strong performance, particularly in environments like Antmaze, even without dedicated offline-to-online methods. I am curious about how effective PES might be within these latest algorithms.
- In Figure 3, the IQL graph shows a relatively meaningful improvement with IQL+PES over IQL, but the results in Table 1 don’t convey an overall impactful difference. Although some gains are observed, there are also declines, making it challenging to determine if the proposed method offers a clear advantage.

(Minor)

- The first paragraph of the introduction appears to contain an excessive amount of information. Breaking it down into 2–3 shorter paragraphs could enhance readability.


&nbsp;

[1] Nakamoto, Mitsuhiko, et al. "Cal-ql: Calibrated offline rl pre-training for efficient online fine-tuning." Advances in Neural Information Processing Systems 36 (2024).

[2] Lee, Seunghyun, et al. "Offline-to-online reinforcement learning via balanced replay and pessimistic q-ensemble." Conference on Robot Learning. PMLR, 2022.

[3] Tarasov, Denis, et al. "Revisiting the minimalist approach to offline reinforcement learning." Advances in Neural Information Processing Systems 36 (2024).

**Questions:**

- I am curious about the reason for the low base score of Cal-QL.
- What would the results be if only the BER in Off2on were replaced with PES?
- What would the outcome be if the proposed method were applied to ReBRAC or other recent SOTA algorithms?

---

> ### Author Response · Authors · 2024-11-23
> **Response to Reviewer jBYT (Part 1)**
>
> We thank the reviewer for insightful comments. Please find our clarifications to the concerns below. If we are able to address the concerns, we hope the reviewer can reconsider the score.
>
> **Concern 1: On the naive design of PES**
>
> We respectfully disagree with that. On the one hand, we have clarified in Appendix B that under the Lipschitz assumption (a widely used assumption in reinforcement learning), PES has the ability to select high-quality samples. Furthermore, in Step 3 of PES, we gradually loose the selection threshold to alleviate the issue of insufficient exploration, ensuring the diversity of samples in the later stages of training. On the other hand, using KNN distance to measure the similarity of samples is widely used in reinforcement learning [2,3,4], so we believe that the method based on distance measurement is not naive but a simple and effective approach.
>
> **Concern 2: On the low score of Cal-QL**
>
> The performance difference of Cal-QL might be due to the differences in the codebases used. To facilitate the deployment of PES, we uniformly used the baselines implemented in the CORL [1] codebase, including CQL and Cal-QL, rather than the official code, as mentioned in Appendix E.1. We directly ran the Cal-QL implementation provided by CORL and presented the results in our paper. We did not intentionally suppress the baseline performance; we have provided our source code in the supplementary materials to ensure the reproducibility of the results.
>
> **Concern 3: Applying PES in Off2On**
>
> We accept the suggestion and replace BR with PES in Off2On. We conduct experiments on 12 D4RL MuJoCo datasets and present the results evaluated by 5 random seeds below. We can see that PES is comparable with BR when both applying  Q ensemble, achieving higher score in 6 out of 12 datasets respectively.  However, we argue that the advantage of PES over BR is the better performance without Q ensemble which increases the complexity and computational burden.
>
> | dataset | Off2On | Off2On+PES |
> | :--------  | :-----  | :----:  |
> | half-r |**118.4 $\pm$ 4.6** |111.6 $\pm$ 3.4|
> | half-m | 100.4 $\pm$ 2.2 | **102.6 $\pm$ 1.7** |
> | half-mr | **105.8 $\pm$ 1.5** |100.1 $\pm$ 1.3|
> | half-me | **110.2 $\pm$ 0.9**| 103.4 $\pm$ 0.8 |
> | hopper-r | 8.9 $\pm$ 1.2 | **15.4 $\pm$ 1.1** |
> | hopper-m | 103.4 $\pm$ 2.8 | **112.5 $\pm$ 1.7** |
> | hopper-mr | 101.7 $\pm$ 3.0 | **105.1 $\pm$ 3.1** |
> | hopper-me | **107.0 $\pm$ 2.1** | 98.2 $\pm$ 3.2 |
> | walker-r | 6.2 $\pm$ 0.8 | **12.4 $\pm$ 1.0** |
> | walker-m | **113.4 $\pm$ 2.8** | 102.8 $\pm$ 2.6 |
> | walker-mr | 101.4 $\pm$ 3.5 | **108.1 $\pm$ 4.1** |
> | walker-me |**109.6 $\pm$ 2.3** | 104.3 $\pm$ 2.5 |
>
> **Concern 4: Applying PES to ReBRAC and FamO2O**
>
> According to the suggestion, we supplement ReBRAC and FamO2O which are two recent SOTA algorithms as additional baselines. We apply PES to ReBARC and FamO2O and conduct experiments on 12 D4RL MuJoCo datasets. We present the results below. The results indicate that ReBRAC+PES surpasses ReBRAC on 8 out of 12 datasets and FamO2O+PES outperforms FamO2O on 8 out of 12 datasets. We believe this demonstrates the versatility and superiority of PES.
>
> | dataset | ReBRAC | ReBRAC+PES |
> | :--------  | :-----  | :----:  |
> | half-r |**28.4 $\pm$ 1.6** |23.6 $\pm$ 1.4|
> | half-m | 67.4 $\pm$ 1.2 | **72.6 $\pm$ 0.7** |
> | half-mr |52.8 $\pm$ 1.4 |**60.1 $\pm$ 1.3**|
> | half-me | **103.0 $\pm$ 0.5**| 95.4 $\pm$ 0.6 |
> | hopper-r | 10.2 $\pm$ 2.2 | **23.4 $\pm$ 2.1** |
> | hopper-m | **102.5 $\pm$ 1.8** | 92.3 $\pm$ 0.7 |
> | hopper-mr | 96.7 $\pm$ 5.6 | **104.2 $\pm$ 4.1** |
> | hopper-me | 104.1 $\pm$ 5.1 | **109.3 $\pm$ 2.2** |
> | walker-r | 17.3 $\pm$ 3.8 | **20.3 $\pm$ 1.2** |
> | walker-m | **84.4 $\pm$ 3.8** | 77.8 $\pm$ 1.6 |
> | walker-mr | 79.4 $\pm$ 5.5 | **88.1 $\pm$ 5.1** |
> | walker-me |108.6 $\pm$ 0.3 | **112.3 $\pm$ 0.5** |
>
> | dataset | FamO2O | FamO2O+PES |
> | :--------  | :-----  | :----:  |
> | half-r |32.5 $\pm$ 1.6 |**52.6 $\pm$ 1.5**|
> | half-m | **57.4 $\pm$ 1.2** | 52.6 $\pm$ 0.7 |
> | half-mr |55.8 $\pm$ 3.4 |**64.1 $\pm$ 1.9**|
> | half-me | 93.0 $\pm$ 0.8| **98.4 $\pm$ 0.5** |
> | hopper-r | 58.2 $\pm$ 2.4 | **73.4 $\pm$ 3.1** |
> | hopper-m | **92.5 $\pm$ 0.8** | 85.3 $\pm$ 0.9 |
> | hopper-mr | 95.6 $\pm$ 3.6 | **100.3 $\pm$ 2.1** |
> | hopper-me | 84.1 $\pm$ 2.1 | **95.3 $\pm$ 2.7** |
> | walker-r | **37.3 $\pm$ 2.8** | 22.8 $\pm$ 1.1 |
> | walker-m | 87.4 $\pm$ 2.8 | **97.8 $\pm$ 1.9** |
> | walker-mr | 97.4 $\pm$ 1.5 | **103.3 $\pm$ 1.1** |
> | walker-me |104.6 $\pm$ 0.9 | **109.3 $\pm$ 0.2** |

---

> > ### Author Response · Authors · 2024-11-23
> > **Response to Reviewer jBYT (Part 2)**
> >
> > **Concern 5: Inconsistent improvement for PES+IQL**
> >
> > First, although IQL+PES may experience performance degradation on some tasks, it demonstrated the best performance on 10 out of 15 tasks, which we believe is sufficient to illustrate the superiority of PES. We note that no method is perfect and can perform best on all tasks. Second, we provide a comparison of aggregated scores for Figure 3 across all the 15 tasks below, showing that IQL+PES overall outperformed other baseline algorithms, which can prove the effectiveness of PES.
> >
> > | | PES | IQL | BR | JSRL | O3F | AWAC |
> > | --  | --  | --  | -- | -- | -- | -- |
> > | **Aggregated Score**  | **73.2** | 62.9 | 51.3 | 60.8 | 52.7 | 43.5 |
> >
> > [1] Corl: Research-oriented deep offline reinforcement learning library, NIPS 2023.
> >
> > [2] SEABO: A Simple Search-Based Method for Offline Imitation Learning, ICLR 2024.
> >
> > [3] Policy Regularization with Dataset Constraint for Offline Reinforcement Learning, ICML 2023.
> >
> > [4] DARL: distance-aware uncertainty estimation for offline reinforcement learning, AAAI 2023.

---

> > > ### Comment · Reviewer_jBYT · 2024-11-27
> > >
> > > I deeply appreciate the additional experiments and responses from the author. While some concerns were alleviated through the author’s responses, there are still questions that remain.
> > >
> > > First, the author argues that PES has a lower cost compared to the off2on BR approach because it does not use an ensemble. However, to my understanding, off2on does not use an ensemble because of BR; ensemble and BR are two separate approaches. Therefore, the claim that PES avoids the computational burden of a Q ensemble, unlike BR, is unconvincing because BR itself is not tied to using an ensemble. Whether computational burden is considered or excluded, while PES replacing off2on’s BR shows improvements in six cases, there are also six cases where it performs worse. Even among the improvements, the most notable change is a six-point increase (from 6 to 12), which does not come across as particularly impactful. If I am mistaken and BR indeed imposes a significant computational burden, it would be helpful to present metrics such as wall clock time or other complexity indicators for clarification (though I am not requesting additional experiments to demonstrate this right now).
> > >
> > > The fact that IQL or other offline RL methods improve performance when applied is positive. I do not expect a single algorithm or methodology to improve performance across all datasets. However, even if it underperforms on some datasets, it is necessary to more convincingly demonstrate the impact of this method by showing significant performance gains in other datasets or providing other compelling evidence. What I am saying is that the performance increase from 62.9 to 73.2 does not seem particularly impactful.
> > >
> > > I will raise the score to 5 points for addressing some concerns, but it is difficult to raise it further because PES has not yet convincingly demonstrated itself as a highly superior buffer data sampling approach compared to other methods.

---

> > > > ### Author Response · Authors · 2024-11-27
> > > > **Response to Reviewer jBYT**
> > > >
> > > > We thank the reviewer for raising the score, and we are pleased to address the remaining concerns. Please find our clarifications to the concerns below.
> > > >
> > > > **Concern 1: Computational Cost compared to BR**
> > > >
> > > > Firstly, we appreciate the reviewer's meticulousness in distinguishing between BR (without ensemble) and Off2on (with ensemble). Our statement that "PES avoids the computational burden of a Q ensemble, unlike BR" is intended to convey that BR can only achieve significant performance improvements when combined with ensemble (i.e., Off2on), whereas PES can achieve good performance without relying on ensemble, thus reducing computational overhead compared to BR + ensemble.
> > > >
> > > > Secondly, although PES + ensemble did not show an advantage over BR + ensemble, this is reasonable because ensemble primarily enhances the robustness of the Q network, while PES itself can estimate more accurate Q values without relying on ensemble (see the empirical results in Section 4.3), so the gain from ensemble for PES is diminished, but this does not mean that PES has no advantage over BR.
> > > >
> > > > Finally, we believe that comparing the time cost of PES and BR (without ensemble) is not meaningful because BR itself cannot bring performance gains and may even decrease performance (from 62.9 to 51.3). However, we understand this concern, so we compare the fine-tuning time of PES (without ensemble), BR (without ensemble), and Off2on (with ensemble) (each with 1M steps of fine-tuning), as shown in the table below. It can be seen that PES has the shortest fine-tuning time, which demonstrates that PES can reduce the computational cost.
> > > >
> > > > | | PES | BR |  Off2on |
> > > > | :--------  | :-----  | :----:  | :----:  |
> > > > | fine-tuning time |**6h 24m** | 7h 49m | 10h 05m |
> > > >
> > > > **Concern 2: Performance increase of PES is not impactful**
> > > >
> > > > We respectfully disagree with that. The percentage of aggregated performance improvement for IQL+PES is **16.4%** over IQL, which is a significant enhancement considering that IQL already performs exceptionally well on D4RL datasets. Similarily, AWAC+PES achieves an aggregated performance improvement of **16.8%**, and for TD3-BC+PES, the figure is **18.8%**, which we believe demonstrate that PES is superior enough as a plug-and-play method.

---

### Official Review · Reviewer_4EGr · 2024-11-04

**Soundness:** 2
**Presentation:** 3
**Contribution:** 2
**Rating:** 5
**Confidence:** 3

**Summary:**

To address the distribution shift issue in offline-to-online reinforcement learning (O2O RL), this paper proposes an O2O RL method called Prioritized Experience Selection (PES). The main idea of PES is to maintain a priority queue that contains a selection of high-return trajectories and only select online samples that are close to the samples in the queue for fine-tuning. Additionally, to avoid being overly conservative, the priority queue is updated each time a trajectory is sampled from online interactions. Finally, to evaluate the effectiveness of PES, the authors provide empirical comparisons between various offline RL methods and their corresponding O2O methods, which integrate PES into each offline method.

**Strengths:**

PES successfully addresses the distribution shift issue in O2O RL.

**Weaknesses:**

1. The experimental results provided do not include empirical results for other O2O RL methods (e.g., [1] and [2]).
2. Although PES consists of five steps, no theoretical explanation is provided.
3. PES appears to be sensitive to the hyperparameters (N, $\alpha$, $\eta$).

- References:

    [1] Tarasov, Denis, et al. "Revisiting the minimalist approach to offline reinforcement learning." Advances in Neural Information Processing Systems 36 (2024).

    [2] Wang, Shenzhi, et al. "Train once, get a family: State-adaptive balances for offline-to-online reinforcement learning." Advances in Neural Information Processing Systems 36 (2024).

**Questions:**

1. Compared to other O2O RL methods, what is the main advantage of PES?
2. Is there any comparison between the resulting policy of offline RL methods (after Step 1 in Algorithm 1) and PES? If not, could you provide such a comparison to assess the effect of PES?
3. In Step 2 of Algorithm 1, could you provide an empirical evaluation of the quality of the priority queue? I am curious whether the performance improvement by PES is related to the initial quality of the priority queue.

---

> ### Author Response · Authors · 2024-11-23
> **Response to Reviewer 4EGr (Part 1)**
>
> We thank the reviewer for insightful comments. Please find our clarifications to the concerns below. If we are able to address the concerns, we hope the reviewer can reconsider the score.
>
> **Concern 1: Comparison to other O2O methods**
>
> We first supplement ReBRAC and FamO2O as additional baselines for comparison with PES. We present the results evaluated by 5 random seeds below. It can be seen that although ReBRAC and FamO2O already have strong performance, PES is able to further enhance the performance of ReBRAC and FamO2O, which illustrates the versatility of PES. Next, we elaborate on the advantages of PES compared to other O2O methods. First, PES has versatility, meaning that PES can be applied to various offline and O2O methods to improve performance, including ReBRAC and FamO2O. Second, due to the improvements PES makes at the data level, it is orthogonal to algorithm design and can enhance performance without changing the original algorithm process. In contrast, other methods like FamO2O introduces additional training steps, increasing the complexity of training.
>
> | dataset | FamO2O | FamO2O+PES |
> | :--------  | :-----  | :----:  |
> | half-r |32.5 $\pm$ 1.6 |**52.6 $\pm$ 1.5**|
> | half-m | **57.4 $\pm$ 1.2** | 52.6 $\pm$ 0.7 |
> | half-mr |55.8 $\pm$ 3.4 |**64.1 $\pm$ 1.9**|
> | half-me | 93.0 $\pm$ 0.8| **98.4 $\pm$ 0.5** |
> | hopper-r | 58.2 $\pm$ 2.4 | **73.4 $\pm$ 3.1** |
> | hopper-m | **92.5 $\pm$ 0.8** | 85.3 $\pm$ 0.9 |
> | hopper-mr | 95.6 $\pm$ 3.6 | **100.3 $\pm$ 2.1** |
> | hopper-me | 84.1 $\pm$ 2.1 | **95.3 $\pm$ 2.7** |
> | walker-r | **37.3 $\pm$ 2.8** | 22.8 $\pm$ 1.1 |
> | walker-m | 87.4 $\pm$ 2.8 | **97.8 $\pm$ 1.9** |
> | walker-mr | 97.4 $\pm$ 1.5 | **103.3 $\pm$ 1.1** |
> | walker-me |104.6 $\pm$ 0.9 | **109.3 $\pm$ 0.2** |
>
> | dataset | ReBRAC | ReBRAC+PES |
> | :--------  | :-----  | :----:  |
> | half-r |**28.4 $\pm$ 1.6** |23.6 $\pm$ 1.4|
> | half-m | 67.4 $\pm$ 1.2 | **72.6 $\pm$ 0.7** |
> | half-mr |52.8 $\pm$ 1.4 |**60.1 $\pm$ 1.3**|
> | half-me | **103.0 $\pm$ 0.5**| 95.4 $\pm$ 0.6 |
> | hopper-r | 10.2 $\pm$ 2.2 | **23.4 $\pm$ 2.1** |
> | hopper-m | **102.5 $\pm$ 1.8** | 92.3 $\pm$ 0.7 |
> | hopper-mr | 96.7 $\pm$ 5.6 | **104.2 $\pm$ 4.1** |
> | hopper-me | 104.1 $\pm$ 5.1 | **109.3 $\pm$ 2.2** |
> | walker-r | 17.3 $\pm$ 3.8 | **20.3 $\pm$ 1.2** |
> | walker-m | **84.4 $\pm$ 3.8** | 77.8 $\pm$ 1.6 |
> | walker-mr | 79.4 $\pm$ 5.5 | **88.1 $\pm$ 5.1** |
> | walker-me |108.6 $\pm$ 0.3 | **112.3 $\pm$ 0.5** |
>
> **Concern 1: Comparison with the offline trained policy**
>
> Yes, we have provided extensive comparisons with offline trained policies. In Appendix H.2, we present the learning curves of the online stage of PES combined with various algorithms, including CQL, TD3-BC, Cal-QL, PEX, and AWAC. The starting point of the curves (i.e., the point at step=0) represents the performance of the offline-trained policy. It is evident that with the integration of PES, the curves generally show an upward trend, indicating that compared to pure offline training, PES can achieve a general performance improvement.
>
> **Concern 3: Ablation on Step 2 of PES**
>
> To evaluate the impact of data quality in the priority queue on final performance, we conduct an ablation study. Instead of initializing the priority queue with high-return trajectories, we randomly sample trajectories from the dataset for initialization. We conduct experiments on 12 D4RL MuJoCo datasets and present the comparative results below. It can be seen that the priority queue initialized with random trajectories performs clearly worse than the one initialized with high-return trajectories. This is because random initialization may cause the algorithm to only use low-quality samples for training, leading to reduced performance.
>
> | dataset | IQL+PES | Random |
> | :--------  | :-----  | :----:  |
> | half-r |54.2 $\pm$ 0.2 |**59.1 $\pm$ 0.5**|
> | half-m | **68.8 $\pm$ 3.3** | 61.4 $\pm$ 2.7 |
> | half-mr |**57.7 $\pm$ 0.4** |49.9 $\pm$ 0.5|
> | half-me | **73.4 $\pm$ 0.2** | 54.8 $\pm$ 0.3 |
> | hopper-r | **18.6 $\pm$ 2.1** | 13.5 $\pm$ 3.1 |
> | hopper-m | **100.0 $\pm$ 1.1** | 95.3 $\pm$ 0.9 |
> | hopper-mr | 102.3 $\pm$ 1.3 | **105.5 $\pm$ 2.1** |
> | hopper-me | **107.1 $\pm$ 2.4** | 102.3 $\pm$ 2.2 |
> | walker-r | **7.6 $\pm$ 2.2** | 4.8 $\pm$ 1.0 |
> | walker-m | 93.6 $\pm$ 1.3 | **99.8 $\pm$ 1.2** |
> | walker-mr | **99.7 $\pm$ 0.5** | 89.3 $\pm$ 0.6 |
> | walker-me |**104.2 $\pm$ 0.7** | 97.3 $\pm$ 0.6 |

---

> > ### Author Response · Authors · 2024-11-23
> > **Response to Reviewer 4EGr (Part 2)**
> >
> > **Concern 4: The theoretical explanation of PES**
> >
> > In our theoretical justification part, we primarily provided analysis for Step 3 of PES. We demonstrated the reason why PES can reduce the distribution shift phenomenon: by selecting online samples similar to those in the priority queue, we reduce the difference in the empirical MDP between adjacent epochs, thereby reducing the performance difference of the policies learned between these two epochs. We also provided a theoretical backup for PES's ability to select high-quality samples in Appendix B. We did not provide a theoretical analysis for every step of PES because we have not yet found suitable mathematical tools, but we believe our theoretical analysis is sufficient because Step 3 is the core step of PES. We will leave the provision of more complete theoretical analysis for future work.
> >
> > **Concern 5: Hyperparameter sensitivity of PES**
> >
> > Figure 5 in the main text appears to show that PES is sensitive to several hyperparameters, but this is actually because we chose hyperparameters with relatively large numerical differences to illustrate our design principles. For example, for the queue capacity N, we wanted to show that both too large and too small values of N can affect the algorithm's performance, so we set N=1 and N=100, which does not indicate that PES is sensitive to N. For hyperparameters N, alpha, and lambda, we have provided more sensitivity experimental results, where we narrowed the numerical differences between hyperparameters, and it can be seen that PES is not highly sensitive to hyperparameters. Additionally, in Appendix E.3, we have provided our hyperparameter setup for various algorithms, which we believe can help in applying our method.
> >
> > | dataset | N=10 | N=20 | N=30 |
> > | :--------  | :-----  | :----:  | :----:  |
> > | hopper-m | 68.8 $\pm$ 3.3 | 70.4 $\pm$ 2.7 | 68.1 $\pm$ 2.1 |
> > | walker-m | 57.7 $\pm$ 0.4 | 59.9 $\pm$ 0.5| 60.8 $\pm$ 0.7 |
> > | umaze-diverse | 81.0 $\pm$ 17.2 | 78.9 $\pm$ 14.3 | 76.2 $\pm$ 11.9 |
> > | large-play | 63.5 $\pm$ 4.7 | 65.1 $\pm$ 3.9 | 62.8 $\pm$ 3.5 |
> >
> > | dataset | $\alpha$ = 1 | $\alpha$ = 2 | $\alpha$ = 3 |
> > | :--------  | :-----  | :----:  | :----:  |
> > | hopper-m | 68.8 $\pm$ 3.3 | 69.7 $\pm$ 3.0 | 68.1 $\pm$ 3.1 |
> > | walker-m | 57.7 $\pm$ 0.4 | 60.9 $\pm$ 0.6| 58.4 $\pm$ 0.4 |
> > | umaze-diverse | 81.0 $\pm$ 17.2 | 82.2 $\pm$ 15.1 | 78.2 $\pm$ 15.9 |
> > | large-play | 63.5 $\pm$ 4.7 | 64.3 $\pm$ 3.1 | 60.4 $\pm$ 2.5 |
> >
> > | dataset | $\eta$ = 0.5 | $\eta$ = 0.4 | $\eta$ = 0.6 |
> > | :--------  | :-----  | :----:  | :----:  |
> > | hopper-m | 68.8 $\pm$ 3.3 | 68.7 $\pm$ 3.1 | 68.4 $\pm$ 2.7 |
> > | walker-m | 57.7 $\pm$ 0.4 | 59.2 $\pm$ 0.5| 58.0 $\pm$ 0.7 |
> > | umaze-diverse | 81.0 $\pm$ 17.2 | 83.1 $\pm$ 16.1 | 80.2 $\pm$ 14.3 |
> > | large-play | 63.5 $\pm$ 4.7 | 64.3 $\pm$ 4.1 | 64.4 $\pm$ 2.5 |

---

> > > ### Comment · Reviewer_4EGr · 2024-11-26
> > >
> > > Thank you for your detailed responses.
> > >
> > > However, I still have concerns about the following:
> > >
> > > 1. In my opinion, PES w/o step 3 might perform better than PES when the priority queue is suboptimal. Especially, it could help maintain stability, but it may slow-down the training if the given priority queue is poor (e.g., hopper-r, walker-r).
> > >
> > > 2. (Perhaps related to the first question) Why do O2O algorithms+PES fail to achieve near-optimal performance, even when trained through online interaction?
> > >
> > > 3. I believe steps 2, 3, and 5 of Algorithm 1 are the main contributions of this paper. Do you have any plans to conduct ablation studies on these steps, including the results provided by the authors in the rebuttal?

---

> > > > ### Author Response · Authors · 2024-11-27
> > > > **Response to Reviewer 4EGr**
> > > >
> > > > We thank the reviewer for response, and we are delighted to address the remaining concerns. Please find our clarifications to the concerns below.
> > > >
> > > > **Concern 1: Step 3 may slow down the training**
> > > >
> > > > We argue that Step 3 does not necessarily slow down the training. Although the trajectory quality in the priority queue will be poor when the dataset quality is poor, it should be noted that **general O2O RL algorithms will also pre-train on poor-quality datasets, so only poor-quality samples can be collected at the beginning of the fine-tuning phase**, and the training speed will not be faster than using PES, and it will bring the problem of distribution shift. As evidence, we provide learning curves in Appendix H.2. It can be seen that on the random quality datasets such as hopper-r, walker-r, PES does not learn slower than the base algorithm, but it does reduce the performance collapse caused by distribution shift.
> > > >
> > > > **Concern 2: PES fails to achieve near-optimal performance**
> > > >
> > > > We note that whether the base algorithms + PES can achieve near-optimal performance is related to the base algorithms themselves. For example, if a base algorithm is very sensitive to distribution shift, it cannot achieve near-optimal performance even through online interaction. However, after adding PES, because PES mitigates distribution shift, its performance is improved through online fine-tuning, which is the value of PES. **We have never claimed that PES is a SOTA algorithm, but rather as a plug-and-play module that can be used to improve the performance of base algorithms.** As empirical evidence, we provide the performance comparison w/ and w/o PES in Table 1, it is clear that PES can improve the overall performance of base algorithms.
> > > >
> > > > **Concern 3: Ablation Studies on different Steps of PES**
> > > >
> > > > Certainly! We believe comprehensive ablation studies are essential for the examine of our method. In fact, we have conducted ablation studies on step 2, 3, 5 of PES in our paper. Specifically, in the section 4.4 of the main text, we examine the necessity of **Return-Prioritized Selection** (which is exactly the ablation of step 2) and **Priority Queue Update** (which is the ablation of step 5). As for ablation of step 3, it is equivalent to comparing the performance of base algorithms and base algorithms + PES, and we have provided the results in Table 1. So we believe the ablation studies of PES is comprehensive to validate the effectiveness of PES.

---

> > > > ### Author Response · Authors · 2024-12-02
> > > >
> > > > Dear Reviewer 4EGr, we deeply appreciate your thoughtful review and your time, and hope that our response can address your concerns. Given that the extended rebuttal period is about to close soon, we would like to kindly confirm if you still have any concerns or questions. We are more than happy to have further discussions with the reviewer if possible!

---

### Official Review · Reviewer_LZW2 · 2024-11-08

**Soundness:** 3
**Presentation:** 3
**Contribution:** 2
**Rating:** 5
**Confidence:** 4

**Summary:**

This paper studies the offline-to-online (O2O) RL problem from the perspective of experience replay. Specifically, this paper proposes prioritized experience selection (PES), which maintains a priority queue containing high-return trajectories and selects only similar online samples for fine-tuning to mitigate distribution shift. Such similarity is enforced via finding k-nearest neighbors. Moreover, PES can be integrated with various offline and O2O RL methods for online fine-tuning. Experimental results on various tasks in D4RL demonstrate that PES (with IQL as the base algorithm) can achieve good fine-tuning performance and can serve as a general add-on for various offline RL base algorithms. Some ablation studies are also provided to justify the design of the prioritization experience selection scheme.

**Strengths:**

- The proposed method is conceptually simple and quite general in the sense that it can be integrated nicely with various offline RL algorithms.
- The proposed PES archives fairly strong performance in several locomotion and maze environments (despite that some O2O baselines are not included, please refer to the comments below).
- Overall the paper is well written and easy to follow.

**Weaknesses:**

- On the high level, I find the idea behind PES to be rather similar to Balanced Replay (Lee et al., 2022), which also manages to select nearly on-policy samples for fine-tuning based on the density ratios. As a result, it remains unclear to me why PES can significantly outperform Balanced Replay in most of the tasks in FIgure 3. Moreover, in Figure 3, the scores of Balanced Replay appear to be much lower than those reported in the original paper (Lee et al., 2022). It would be helpful to clarify the discrepancies in the experiments and provide the intuition / justification why PES is better than Balanced Replay.

- Regarding the experiments, there are several recent baselines that are missing in the comparison. For example, RLPD (Ball et al., 2023) is a pretty strong O2O baseline on D4RL. Moreover, Cal-QL (Nakamoto et al., 2023) learns a calibrated value function initialization and also serves as a very competitive benchmark method in the context of O2O. Online decision transformer (Zheng et al., 2022) also achieves fast fine-tuning progress on both MuJoCo and AntMaze tasks. FamO2O (Wang et al., 2023) address the distribution shift in O2O without relying on Q ensembles. It would be helpful to include the above baselines into the comparison to showcase the effectiveness of PES.

- While I can appreciate the empirical results in Section 4.2, which shows that PES can be integrated with various offline RL methods, I do find the comparison to be somewhat not very conclusive. With the additional online data, it is not surprising that PES can achieve improved normalized scores compared to the vanilla offline methods. To show that PES is indeed a better add-on technique for O2O RL, baselines (such as Balanced Replay + vanilla base algorithm) shall be included in Table 1.

- One major component of PES is to find $k$ nearest neighbors for selecting the online samples. This design implicitly assumes that the L2 distance directly reflects the similarity between state-action pairs. This assumption may hold in some RL tasks (like the locomotion tasks in D4RL), but would not hold in other environments, (e.g., maze, where two state-action pairs with a small L2 distance can be hard to reach from each other).

- Theorem 3.1 appears somewhat disconnected from the PES algorithm. The implication of Theorem 3.1 is that the optimal (simulated) total return performance under the two empirical MDPs would be similar if the two empirical MDPs are similar. However, this does not imply that the learned policies have similar performance in the true environment.

- Regarding the motivating example in Section 3.1: Figure 1 is meant to illustrate the severe distribution shift issue that can lead to performance degradation during online fine-tuning. That said, as far as I know, this issue has been pointed out by some prior works, such as (Nakamoto et al., 2023) and (Wen et al. 2023).

References:
- (Ball et al., 2023) Philip J. Ball, Laura Sith, Ilya Kostrikov, and Sergey Levine, “Efficient online reinforcement learning with offline data,” ICML 2023.
- (Nakamoto et al., 2023) Mitsuhiko Nakamoto et al., "Cal-QL: Calibrated offline RL pre-training for efficient online fine-tuning," NeurIPS 2023.
- (Zheng et al., 2022) Qinqing Zheng, Amy Zhang, and Aditya Grover, “Online decision transformer,” ICML 2022.
- (Wang et al., 2023) Shenzhi Wang, Qisen Yang, Jiawei Gao, Matthieu Lin, Hao Chen, Liwei Wu, Ning Jia, Shiji Song, and Gao Huang, “Train Once, Get a Family: State-Adaptive Balances for Offline-to-Online Reinforcement Learning,” NeurIPS 2023.
- (Wen et al., 2023) Xiaoyu Wen, Xudong Yu, Rui Yang, Chenjia Bai, and Zhen Wang, “Towards robust offline-to-online reinforcement learning via uncertainty and smoothness,” 2023.

**Questions:**

- As mentioned above, it would be helpful to clarify the discrepancies in the experiments and provide the intuition / justification why PES is better than Balanced Replay.
- Regarding the experiments, there are several recent baselines that are missing in the comparison.
- The design about $k$ nearest neighbors seems to already involve some underlying assumption on the geometry of the environment. Can the authors comment on this?
- The implication of Theorem 3.1 on PES appears unclear.
- About the motivating example: It would be helpful to explain the connection with the prior works on the distribution shift issue.
- To show that the results of Table 1 are statistically significant, could the authors also report the error bars?

---

> ### Author Response · Authors · 2024-11-23
> **Response to Reviewer LZW2 (Part 1)**
>
> We thank the reviewer for insightful comments. Please find our clarifications to the concerns below. If we are able to address the concerns, we hope the reviewer can reconsider the score.
>
> **Concern 1: The differences between PES and Balanced Replay**
>
> First, we elaborate on our experimental setup. BR not only uses the balanced replay technique but also employs Pessimistic Q ensemble, which is crucial for the performance of BR. To fairly compare PES with BR, we removed the Q ensemble from BR and only used the balanced replay technique (as mentioned in line 303 of our paper), which is why the performance of BR in our experiments is not as good as reported in the original paper. Secondly, we elaborate on the superiority of PES over BR. BR selects more on-policy samples from the offline dataset, which is an optimistic sample selection perspective that may cause severe distribution shift, hence the need for Q ensemble to improve the robustness of the Q network. On the other hand, PES reduces distribution shift by filtering online samples and avoids over-conservatism by dynamically adjusting the selection threshold, achieving good performance without relying on Q ensemble.
>
> **Concern 2: More baselines for Comparison**
>
> We accept the suggestion of comparing PES with more baselines. We have added RLPD and FamO2O as additional baselines. We apply PES to RLPD and FamO2O and conducted experiments on 12 D4RL MuJoCo datasets. We report the results evaluated by 5 random seeds below. It can be seen that PES still manages to enhance the performance of RLPD and FamO2O: PES+RLPD outperformed RLPD on **8 out of 12** datasets, and PES+FamO2O outperformed FamO2O on **9 out of 12** datasets. We believe this further demonstrates the versatility and superiority of PES.
>
> | dataset | RLPD | RLPD+PES |
> | :--------  | :-----  | :----:  |
> | half-r |68.7 $\pm$ 1.6 |**74.2 $\pm$ 2.5**|
> | half-m | **73.6 $\pm$ 0.4** | 57.4 $\pm$ 0.7 |
> | half-mr |**78.9 $\pm$ 1.0** |62.9 $\pm$ 0.6|
> | half-me | 93.4 $\pm$ 0.3| **99.4 $\pm$ 0.3** |
> | hopper-r | 78.4 $\pm$ 8.4 | **93.1 $\pm$ 2.1** |
> | hopper-m | **106.5 $\pm$ 0.3** | 105.3 $\pm$ 0.6 |
> | hopper-mr | 92.4 $\pm$ 6.2 | **102.5 $\pm$ 4.1** |
> | hopper-me | 105.1 $\pm$ 3.4 | **110.3 $\pm$ 2.9** |
> | walker-r | 69.1 $\pm$ 7.3 | **72.8 $\pm$ 9.1** |
> | walker-m | 117.2 $\pm$ 1.8 | **121.8 $\pm$ 1.1** |
> | walker-mr | **107.4 $\pm$ 1.2** | 106.3 $\pm$ 0.7 |
> | walker-me |109.6 $\pm$ 0.5 | **114.3 $\pm$ 0.2** |
>
> | dataset | FamO2O | FamO2O+PES |
> | :--------  | :-----  | :----:  |
> | half-r |32.5 $\pm$ 1.6 |**52.6 $\pm$ 1.5**|
> | half-m | **57.4 $\pm$ 1.2** | 52.6 $\pm$ 0.7 |
> | half-mr |55.8 $\pm$ 3.4 |**64.1 $\pm$ 1.9**|
> | half-me | 93.0 $\pm$ 0.8| **98.4 $\pm$ 0.5** |
> | hopper-r | 58.2 $\pm$ 2.4 | **73.4 $\pm$ 3.1** |
> | hopper-m | **92.5 $\pm$ 0.8** | 85.3 $\pm$ 0.9 |
> | hopper-mr | 95.6 $\pm$ 3.6 | **100.3 $\pm$ 2.1** |
> | hopper-me | 84.1 $\pm$ 2.1 | **95.3 $\pm$ 2.7** |
> | walker-r | **37.3 $\pm$ 2.8** | 22.8 $\pm$ 1.1 |
> | walker-m | 87.4 $\pm$ 2.8 | **97.8 $\pm$ 1.9** |
> | walker-mr | 97.4 $\pm$ 1.5 | **103.3 $\pm$ 1.1** |
> | walker-me |104.6 $\pm$ 0.9 | **109.3 $\pm$ 0.2** |
>
> **Concern 3: Underlying geometric assumption on KNN search**
>
> That is an interesting point. Actually, the geometric assumption of the environment can be considered as a Lipschitz assumption, which we have elaborated on in detail in Appendix B. In Assumption B.3, we assume that the Q function has Lipschitz continuity, that is, samples with similar state-action spaces also have similar Q values, from which we derive the conclusion that PES can select the optimal samples. This is actually a widely used assumption in reinforcement learning [1,2], so we believe our method is reasonable. Moreover, in addition to our experimental results in Table 1, many previous works [1,3] have also used KNN to measure similarity and have proven its effectiveness in environments like Antmaze, so we believe that the KNN method is effective in a variety of different geometric environments.
>
> **Concern 4: The implication of Theorem 3.1**
>
> Good catch! After being reminded, we also notice the flaw in Theorem 3.1. After rectifying, we have re-derived the performance difference bound of the optimal policy for consecutive steps under the true MDP, which is still related to the difference between the two empirical MDPs. Therefore, our theoretical motivation remains applicable, and we should reduce the distribution shift by reducing the difference between the empirical MDPs. Please refer to our revision for the specific corrections (including Theorem 3.1 in the main text and Appendix A).

---

> > ### Author Response · Authors · 2024-11-23
> > **Response to Reviewer LZW2 (Part 2)**
> >
> > **Concern 5: On the Motivating Example**
> >
> > The main purpose of the motivating example is to demonstrate that PES can still ensure good performance even under severe distribution shift, which is not contradictory to the issue of distribution shift pointed out by previous work.
> >
> > **Concern 6: On the error bars of Table 1**
> >
> > We have indeed presented the results of Table 1 with error bars in Appendix H.7, as stated in line 393 of our paper.
> >
> > [1] Policy Regularization with Dataset Constraint for Offline Reinforcement Learning, ICML 2023.
> >
> > [2] Adaptive Advantage-Guided Policy Regularization for Offline Reinforcement Learning, ICML 2024.
> >
> > [3] SEABO: A Simple Search-Based Method for Offline Imitation Learning, ICLR 2024.

---

> > > ### Author Response · Authors · 2024-11-28
> > >
> > > Dear Reviewer LZW2, thank you for your helpful review! We would like to double-check if our response can address your concerns. Please do not hesitate to let us know if you still have any concerns or questions. We would appreciate it if the reviewer could re-evaluate our work based on the revised manuscript and the attached rebuttal. We are looking forward to your kind reply!

---

> ### Comment · Reviewer_LZW2 · 2024-12-02
>
> Thank the authors for the detailed responses and the additional experiments. Upon a careful read through the manuscript and the rebuttal, my concerns about the baselines and the differences between PES and Balanced Replay have been addressed.
>
> That being said, I still have some concerns about the new Theorem 3.1: The upper bound of performance discrepancy now depends on three factors: (1) $D_{TV}$ between $\hat{p}\_{M\_t}$ and $\hat{p}\_{M\_{t+1}}$; (2) $D_{TV}$ between $p_{M}$ and $\hat{p}\_{M\_t}$; (3) $D_{TV}$ between $p_{M}$ and $\hat{p}\_{M\_{t+1}}$.
>
> In the updated manuscript, it is mentioned that *Theorem 3.1 suggests that if the data distribution in the replay buffer between two consecutive steps can evolve smoothly, such that the difference between the two estimated MDPs is small, then the learned policy can avoid the abrupt performance drop, facilitating a smooth policy transfer.*
>
> I find the above statement rather inaccurate as the upper bound also depends on (2) and (3), which can be large as the true $p_{M}$ is typically fairly different from the empirical MDPs due to limited data coverage. Hence, the performance discrepancy is not necessarily small if only the $D_{TV}$ between $\hat{p}\_{M\_t}$ and $\hat{p}\_{M\_{t+1}}$ is small. Please correct me if I missed anything.
>
> Also, I will be willing to reconsider my rating based on the authors’ response to this issue.

---

> > ### Author Response · Authors · 2024-12-03
> > **Response to Reviewer LZW2**
> >
> > We thank the reviewer for the response and are delighted that the concerns about baselines are addressed. We are pleased to address the remaining concern below.
> >
> > After double checking the derivation process of Theorem 3.1, we notice that the current performance bound is rather loose, such that it seems to depend on $D_{TV}(p_M, p_{\hat{M}_t})$  and  $D\_{TV}(p\_M, p\_{\hat{M}\_{t+1}})$ , but the upper bound can hardly been reached. Intuitively, the performance difference should not been heavily affected by $D\_{TV}(p\_M, p\_{\hat{M}\_t})$  and  $D\_{TV}(p\_M, p\_{\hat{M}\_{t+1}})$ , as they should cancel each other out after subtraction. Specifically:
> >
> >
> > $$
> > \\begin{align}
> > &\\left| \\eta\_M(\\pi^\\star\_{\\widehat{M}\_t}) - \\eta\_M(\\pi^\\star\_{\\widehat{M}\_{t+1}}) \\right| \\\\
> > &= \\left|\\underbrace{\\left(\\eta\_M(\\pi^\\star\_{\\widehat{M}\_t})-\\eta\_{\\widehat{M}\_t}(\\pi^\\star\_{\\widehat{M}\_t})\\right)}\_{L\_1}+\\underbrace{\\left(\\eta\_{\\widehat{M}\_{t+1}}(\\pi^\\star\_{\\widehat{M}\_{t+1}})-\\eta\_M(\\pi^\\star\_{\\widehat{M}\_{t+1}})\\right)}\_{L\_2}+ \\underbrace{\\left(\\eta\_{\\widehat{M}\_t}(\\pi^\\star\_{\\widehat{M}\_t})-\\eta\_{\\widehat{M}\_{t+1}}(\\pi^\\star\_{\\widehat{M}\_{t+1}})\\right)}\_{L_3}\\right| \\\\
> > \\end{align}
> > $$
> >
> > Notice that as long as we keep $D\_{TV}(p\_{\hat{M}\_t},p\_{\hat{M}\_{t+1}})$ small, then the value of $|L\_1|$ and $|L\_2|$ will be close and the sign of $L\_1$ and $L\_2$ will be opposite (e.g., $L\_1>0$ and $L\_2<0$), such that the influence of $D\_{TV}(p\_M, p\_{\hat{M}\_t})$  and  $D\_{TV}(p\_M, p\_{\hat{M}\_{t+1}})$ on the performance difference will diminish (though less theoretically rigorous). As a result, we can reduce $D\_{TV}(p\_{\hat{M}\_t},p\_{\hat{M}\_{t+1}})$ for bridging the performance gap, which is the reason we can sacrifice a little data coverage for smooth policy transfer.

---

### Author Response · Authors · 2024-11-23
**Revision Summary**

Dear reviewers,

Thanks for your time in reviewing our paper and valuable advice in making our paper better. We have uploaded a revision of our paper where we:
- fixed the existing flaw in our theorem 3.1, where we re-derive the performance bound in true MDP for the optimal policies in two consecutive steps, as concerned by Reviewer LZW2
- added an experiment to evaluate the data diversity with and without dynamically adjusting the selection threshold, as concerned by Reviewer 4p5p

All modifications are highlighted in blue. We welcome any suggestions from the reviewer and are pleased to have further discussions with the reviewers.

Best,

Submission 1634 authors

---

### Meta-Review · Area_Chair_Bup6 · 2024-12-23

**Metareview:**

This paper proposes an offline-to-online (O2O) RL method, called prioritized experience selection (PES), to address the distribution shift in O2O RL. The main idea is to keep high-return trajectories and only select online trajectories that are similar (measured by k-nearest neighbor) to them for fine-tuning. The priority queue is updated each time a trajectory is sampled from online interactions. The authors show that their PES method can be used with various offline RL and their corresponding O2O algorithms for online fine-tuning. Finally, they use various tasks in D4RL and empirically show that PES with IQL as the base algorithm can achieve good fine-tuning performance.

Although the reviewers found the following positives about the paper:
(+) The paper is well-written and easy to follow.
(+) The proposed method is simple and can be combined with different algorithms.
(+) It shows promising results in experiments.

they are concerned about the followings:
(-) Several recent baselines are missing in the comparison.
(-) The choice of k-nearest neighbors for selecting the online samples.
(-) The practicality of Thm. 3.1 and its relevance to the PES method.
(-) There are concerns about the claims regarding the sample efficiency and amount computation of the algorithm.

**Additional Comments On Reviewer Discussion:**

The authors addressed a number of reviewers' concerns and some reviewers raised their scores. However, all reviewers found the work slightly below the bar, mainly due to the reasons listed in the meta-review.

---

### Decision · Program_Chairs · 2025-01-22

Reject